



# Combining POLDER-3 satellite observations and WRF-Chem numerical simulations to derive biomass burning aerosol properties over the Southeast Atlantic region

Alexandre Siméon[1], Fabien Waquet[1], Jean-Christophe Péré[1], Fabrice Ducos[1], François Thieuleux[1], Fanny
Peers[1], Solène Turquety[2], Isabelle Chiapello[1]

[1]Université de Lille, CNRS, UMR 8518, LOA – Laboratoire d'Optique Atmosphérique, F-59000 Lille, France
[2]LMD/IPSL, Sorbonne Université, ENS, PSL Université, École Polytechnique, Institut Polytechnique de Paris, CNRS, Paris,
France

*Correspondence to*: Alexandre Siméon (alexandre.simeon@univ-lille.fr) and Fabien Waquet (fabien.waquet@univ-lille.fr)

**Abstract.** Aerosol absorption is a key property to assess the radiative impacts of aerosols on climate at both global and regional
scales. The aerosol physico-chemical and optical properties remain not sufficiently constrained in climate models, with
difficulties to properly represent both the aerosol load and their absorption properties in clear and cloudy scenes, especially
for absorbing biomass burning aerosols (BBA). In this study we focus on biomass burning (BB) particle plumes transported
above clouds over the Southeast Atlantic (SEA) region off the southwest coast of Africa, in order to improve the representation

of their physico-chemical and absorption properties. The methodology is based on aerosol regional numerical simulations from
the WRF-Chem coupled meteorology-chemistry model combined with a detailed inventory of BB emissions and various sets
of innovative aerosol remote sensing observations, both in clear and cloudy skies from the POLDER-3/PARASOL space
sensor. Current literature indicates that some organic aerosol compounds (OC) called "brown carbon" (BrOC), primarily
emitted by biomass combustion absorb the ultraviolet-blue radiation more efficiently than pure black carbon (BC). We exploit

this specificity by comparing the spectral dependence of the aerosol single scattering albedo (SSA) derived from the POLDER-
3 satellite observations in the 443-1020 nm wavelength range with the SSA simulated for different proportions of BC, OC and
BrOC at the source level, considering the homogeneous internal mixing state of particles. These numerical simulation
experiments are based on two main constraints: maintaining a realistic aerosol optical depth both in clear and above cloudy
scenes and a realistic BC/OC mass ratio. Modelling experiments are presented and discussed to link the chemical composition

with the absorption properties of BBA and to provide estimates of the relative proportions of black, organic and brown carbon
in the African BBA plumes transported over the SEA region for July 2008. The absorbing fraction of organic aerosols in the
BBA plumes, i.e., BrOC, is estimated at 2 to 3 %. The simulated mean SSA are 0.81 (565 nm) and 0.84 (550 nm) in clear and
above cloudy scenes respectively, in good agreement with those retrieved by POLDER-3 (0.85±0.05 at 565 nm in clear-sky
and at 550 nm above clouds) for the studied period.



## 1 Introduction

The SEA region which includes the Southeastern Atlantic Ocean (SEAO) and the western and southern parts of Africa is an excellent natural laboratory to better understand the complexity of aerosol-cloud interactions (Keil and Haywood 2003; Stier et al. 2013; Peers et al. 2016; Zuidema et al. 2016a,b). Africa accounts for about half of the global annual carbon emissions produced by BB (Giglio et al. 2003; Giglio, 2006; Reid et al. 2009; Giglio and Randerson, 2010; Werf et al. 2010). These BB,

mostly large-scale man-made (e.g. deforestation, agricultural, domestic practices), are frequent and occur every year mainly during the dry season from June to October in central, eastern and southern Africa (Roberts et al. 2009). BBA are then regularly emitted from these source areas and transported over the SEAO above one of the three largest persistent stratocumulus layers on the planet (Costantino and Bréon, 2013). This long-distance transport is favoured by the anticyclonic circulation which is dominant during the dry season, and induces the transport of air masses westward on its northern periphery (Cahoon et al.

1992; Garstang et al. 1996; Swap et al. 1996).

The presence of absorbing aerosols (AA) above clouds decreases the amount of light reflected by clouds into space, thus causing a reduction of the planet's albedo and a positive regional radiative forcing (warming). Significant positive radiative forcing have been measured in this region (de Graaf et al. 2012, 2014, 2020; Peers et al. 2015) but strong differences persist between simulations of climate models, resulting in high uncertainties in estimating their radiative forcing, especially in this

part of the world (Boucher et al. 2013; Myhre et al. 2013b; Stier et al. 2013). The BBA direct radiative forcing estimated at the top of the atmosphere (TOA) by the sixteen global models of the AeroCom project (Huneeus et al. 2011; Myhre et al. 2013a; Bian et al. 2017) is highly variable with large deviations observed within the range of these climate numerical models: from $-1.16 \, \mathrm{W.m^{-2}}$ for the GMI MERRA model which translates a cooling effect to $+1.62 \, \mathrm{W.m^{-2}}$ for the CAM5.1 MAM3 model reflecting an atmospheric warming effect (Zuidema et al. 2016b). The absorption of solar radiation by BBA above

clouds causes a warming where the aerosol layer is located. This warming would alter the thermodynamic properties of the atmosphere, which would impact the vertical development of low clouds. Clouds under these BBA layers would then be optically thicker and the altitude of their cloud top would be lower. This semi-direct effect has been observed off the coasts of Angola and Namibia (Wilcox 2012; Deaconu et al. 2019). Modelling studies have shown that the BBA semi-direct radiative forcing above stratocumulus clouds over the SEA region is primarily negative at the TOA. It was estimated on average at -2.6

$\mathrm{W.m^{-2}}$ over the July-September period by Sakaeda et al. (2011) or at $-30.5 \, \mathrm{W.m^{-2}}$ during few days of study (period 5-10 of August 2016, Gordon et al. 2018). Recent modelling studies have also shown that African BBA could have an indirect radiative effect of $-10.1 \, \mathrm{W.m^{-2}}$ (5-10 of August 2016, Gordon et al. 2018) and $-8.05 \, \mathrm{W.m^{-2}}$ (August $1^{\mathrm{st}}$ to September 30, 2014, Lu et al. 2018) over the SEA region during the dry season. There are sometimes cases of contact by the cloud top or BBA transport in the lower layers impacting the cloud microphysics such as the increase in liquid water path (Costantino and Bréon 2013).

As with semi-direct effects, it should be noted that there is still no consensus on quantifying the indirect effects of BBA over the SEA region.



These disagreements between AeroCom models on the properties of aerosols above clouds are explained by differences in parameterizations on the aerosol injection height, the aerosol lifetime (related to the processes of deposition, removal, …) and on the parameterization of aerosol absorption (Peers et al. 2016). The literature reports a wide range of BC/OC mass ratio

values for African BBA plumes depending on the different types of burned vegetation in Africa (extratropical forests, tropical forests, savannahs, grasslands and crop residues). These carbonaceous aerosol mixing ratios can be highly variable ranging from 0.06 to 0.33 at the level of BBA emission sources (Bond et al. 2004; Liousse et al. 2010; Werf et al. 2010; Akagi et al. 2011) and from 0.11 to 0.24 during the transport of BBA plumes (Ruellan et al. 1999; Formenti et al. 2003; Kirchstetter et al. 2003; Capes et al. 2008). In addition, a wide range of BC spectral refractive indices is reported in the literature (from 1.6 to

2.0 for its real part and from 0.45 to 1.1 for its imaginary part – Bond and Bergstrom 2006; Bond et al. 2013; Liu et al. 2018), which can be explained by the diversity of experimental techniques implemented with their own uncertainties or by not perfectly pure sample collection (i.e., contaminated by other chemical species) (Bond and Bergstrom 2006; Bond et al. 2013). Current uncertainties are also related to combustion conditions, types of burned fuels, BC aging and its fractal morphology (Sorensen 2001; Bond and Bergstrom 2006). The presence of BrOC and its proportion in the BB plumes (Arola 2011; Laskin

et al. 2015), the variability of its spectral refractive index values (Kirchstetter et al. 2004; Alexander et al. 2008; Chen and Bond 2010; Hoffer et al. 2016, 2017; Sumlin et al. 2017) and its atmospheric lifetime (Samset et al. 2014; Wang et al. 2014) prevent accurate representation of this chemical compound in climate models. Uncertainties about emitted amounts of black and primary organic carbon (Giglio and Randerson 2010; Werf et al. 2010; Wiedinmyer et al. 2011; Kaiser et al. 2012; Turquety et al. 2014; Andreae 2019; Pan et al. 2019) are other key elements highly influencing the BBA absorption properties.

In this context, the SEA region has been the subject of several scientific studies and publications focusing on aerosol-radiation and aerosol-cloud interactions and their climate impacts (Wilcox, 2012; Meyer et al. 2013; Graaf et al. 2014; Peers et al. 2016; Zuidema et al. 2016b). International research programs have been in place since the 1990s (SAFARI-1992, Lindesay et al. 1996) and until very recently around Southern Africa. In particular, the SAFARI-2000 field campaign provided a set of airborne and ground measurements of aerosol absorption properties over the South African region including their single

scattering albedo (Swap et al. 2002; Haywood et al. 2003). The SSA of aerosols could thus be estimated at 0.85±0.02 at 550 nm on regional average and during the BB season, confirming the highly absorbing nature of the aerosols emitted from vegetation fires in this region (Leahy et al. 2007). Nevertheless, some questions about the aerosol absorption spectral dependence or its evolution during aerosol transport have not been resolved with these pioneering campaigns (Formenti et al. 2019). SAFARI-2000's observations also showed that these AA could exert a positive direct radiative forcing above clouds

(Keil and Haywood, 2003). More recently, several new observation campaigns have been put in place during the 2016-2018 period to advance the characterization of BBA in this region and to better understand their interactions with clouds and their radiative effects (Zuidema et al. 2016b). These field campaigns have been initiated by various international teams with in particular the American ORACLES (ObseRvations of Aerosols above CLouds and their intEractionS) and British CLARIFY-2017 (CLoud-Aerosol-Radiation Interactions and Forcing Year 2017) projects. The French teams conducted a field campaign

on the Namibian Atlantic coast during the summer of 2017 as part of the ANR AEROCLO-sA project, deploying a set of





ground-based and airborne measurements (Formenti et al. 2019). This coordinated international effort has provided a comprehensive set of ground and airborne measurements of aerosol and cloud properties over Southern Africa and the South Tropical Atlantic region. Innovative satellite observations of aerosols and clouds derived from the POLarization and Directionality of the Earth's Reflectances (POLDER-3) space sensor in clear-sky (Dubovik et al. 2011, 2014) and above clouds

(Waquet et al. 2013a; Waquet et al. 2013b; Peers et al. 2015) and from the Spinning Enhanced Visible and Infrared Imager (SEVIRI) were also recently developed (Peers et al. 2019; 2020) and offer new opportunities (e.g. daily coverage, high spatial and temporal resolutions).

Progress towards more accurate modelling of BBA transported above clouds and their associated radiative effects in the SEA region requires robust representation of their loads and their physico-chemical and absorption properties (Stier et al. 2013;

Zuidema et al. 2016b; Mallet et al. 2020). Here we use the WRF-Chem coupled meteorology-chemistry regional model, which allows to perform simulations at higher spatial and temporal resolutions than current global climate models with a better representation of aerosol processes, to simulate the life cycle of aerosols, their loads and their absorption properties (Grell et al. 2005; Fast et al. 2006; Peckham, 2012; Powers et al. 2017) over the SEA region. Coincident POLDER-3 innovative satellite aerosol retrievals, available both in clear and cloudy conditions, are used to constrain the chemical composition of BBA

simulated with the WRF-Chem model. The methodology consists in adjusting the aerosol absorption properties in WRF-Chem by analysing its sensitivity to different input parameters, i.e., the amount of black and primary organic carbon emitted by biomass combustion at the source level, the BC/OC mass ratio, and the black and brown carbon absorption. This approach leads to an optimized configuration of WRF-Chem for aerosol simulations over the SEA region, with constrained aerosol key parameters (AOD, SSA, and their spectral dependence) above clouds in the model, in order to properly calculate their forcing

and heating rates and subsequently to better quantify their feedbacks on clouds. Because of the availability of POLDER-3 coincident satellite data, our study focuses on July 2008, this month also corresponding to many BB events and to frequent cases of contact of BBA with stratocumulus clouds favourable to direct, semi-direct and indirect radiative effects.

The structure of this article is as follows. Section 2 describes the modelling tool and satellite data as well as the strategy used to improve the representation of BBA distribution and absorption properties in regional climate models. Section 3 reports

results for the meteorology, the aerosol load, size, horizontal and vertical distribution and then focuses on aerosol absorption properties. Section 4 summarizes the main results and conclusions of our study.

## 2 Data and method

### 2.1 WRF-Chem: a regional meteorological model coupled with chemistry

#### 2.1.1 Description

WRF-Chem (Weather Research and Forecasting model coupled with Chemistry) is a regional model whose specificity is to couple atmospheric chemistry with meteorology (Grell et al. 2005; Fast et al. 2006). Particulate pollution influences climate



change because aerosols affect the radiative transfer properties of the atmosphere. Also, climate change can alter the mechanisms of formation, transport, aging and scavenging of atmospheric particles. WRF-Chem captures this complexity. It is a chemistry-transport model coupled with a meso-scale weather model designed for atmospheric research and operational

forecasting applications (Skamarock et al. 2005, 2008). WRF-Chem is used in many types of air quality and climate research and offers a wide range of options for modelling of gas and particulate chemistry (Grell et al. 2004; Grell and Baklanov 2011; Baklanov et al. 2014; Powers et al. 2017). The main physical and chemical options selected for the detailed simulation of atmospheric chemistry with WRF-Chem V3.9.1.1 initial configuration are shown in Table 1.

| Parameterization | Variable namelist | Option | Module | Reference |
|---|---|---|---|---|
| *Physics* | | | | |
| Cloud microphysics | mp_physics | 2 | Lin (Purdue) | Lin et al. (1983) |
| Cloud convection | cu_physics | 5 | Grell-3 | Grell (1993) Grell et Dévényi (2002) |
| Surface pattern | sf_surface_physics | 2 | Noah LSM | Tewari et al. (2004) |
| Planetary boundary layer | bl_pbl_physics | 1 | YSU | Hong et al. (2006) |
| Shortwave radiation | ra_sw_physics | 2 | Goddard | Chou et Suarez (1994) |
| Longwave radiation | ra_lw_physics | 1 | RRTM | Mlawer et al. (1997) |
| *Chemistry* | | | | |
| Chemistry | chem_opt | 112 | MOZCART+KPP | Pfister et al. (2011) |
| Anthropogenic emissions | emiss_opt | 8 | MOZCART | Pfister et al. (2011) |
| Desert dust emissions | dust_opt | 3 | GOCART AFWA | Jones et al. (2010, 2012) |
| Biogenic emissions | bio_emiss_opt | 3 | MEGAN | Guenther et al. (2006) |
| Biomass burning emissions | biomass_burn_opt | 2 | MOZCART | Pfister et al. (2011) |
| Aerosol mixing state | aer_op_opt | 1 | Volume approximation | Ghan et al. (2001) Fast et al. (2006) Barnard et al. (2010) |

**Table 1: WRF-Chem V3.9.1.1 main physical-chemical options selected for the study of the biomass burning aerosol optical properties**
**and their climate forcing over the Southeast Atlantic region.**

We chose the MOZCART chemistry scheme (Pfister et al. 2011) which links the MOZART (Model for OZone And Related chemical Tracers) gas phase chemistry module (Emmons et al. 2010) with the GOCART (Global Ozone Chemistry Aerosol Radiation and Transport) aerosol module (Chin et al. 2000a; Chin et al. 2000b; Ginoux et al. 2001; Chin et al. 2002) because it is well suited for BB chemistry (Emmons et al. 2015; Lassman et al. 2017). MOZCART simulates the life cycle of the main

aerosol types, i.e., black carbon (BC), organic carbon (OC), sulphates ($SO_4^{2-}$), desert dust (DUST) and sea salts (SEAS), as well as a large number of gaseous species (85 species evolving according to 196 chemical reactions including 39 photolysis).





MOZCART calculates their transport, chemical aging (Helfand et Labraga 1988; Allen et al. 1996; Lin et Rood 1996), and their dry (Fuchs et al. 1965; Wesely 1989) and wet (Giorgi et Chameides 1986; Balkanski et al. 1993) depositions.

For meteorological input data, we used the meteorological reanalyses from NCEP (National Centre for Environmental

Prediction) global reanalysis provided by GFS (Global Forecast System) at 1° horizontal spatial resolution updated every six hours (Kalnay et al. 1996).

We chose the homogeneous internal mixing state of particles in our WRF-Chem numerical simulations because this type of mixture is consistent with that considered in the POLDER-3 satellite inversion algorithms used to retrieve aerosol optical properties. The aerosol radiative feedbacks are not taken into account in the present study.

**2.1.2 Configuration**

Our domain of study centred on the SEA region is illustrated in Figure 1 with the black frame of geographic coordinates (20°W-30°E, 39.4°S-10°N). This domain corresponds to the area affected by transport of BBA plumes off the southwest coasts of Africa. We have simulated an extended area of geographic coordinates (34.1°W-45.1°E, 39.4°S-33°N) in order to take into account all desert, marine, biogenic, anthropogenic and BB emissions of aerosols that could impact our region of interest. The

horizontal spatial resolution of the simulations performed with WRF-Chem in this area is 30x30 km². The atmospheric layer is divided into 50 vertical levels which are distributed from the surface up to a pressure level of 50 hPa.

We have simulated the first half of July 2008 (plus 15 days spin-up) to reduce numerical costs because this time period appears to be representative of the whole month: the Above-Cloud Aerosol Optical Depth (ACAOD) and the Above-Cloud Single Scattering Albedo (ACSSA) at 550 nm from the POLDER-3 retrievals are respectively 0.45 and 0.85 over the first half of July

2008 and 0.46 and 0.85 over July 2008 on average over our combined studied areas (black frames in Figure 2). These values confirm significant amounts of AA transported above clouds of the SEAO during this time of the year.

The studied area for the optical properties of BBA is separated into two distinct areas (black frames in Figure 2). The first area (left) of geographic coordinates (12°E-30°E, 10°S-0°S) corresponds to the BBA emission sources over the Southern African continent and will be analysed in clear atmosphere for comparisons with the POLDER-3/GRASP data (Dubovik et al. 2011,

2014). The second area (right) of geographic coordinates (0°E-15°E, 15°S-5°S) corresponds to the transport of BBA plumes above clouds over the SEAO and will be analysed in cloudy atmosphere for comparisons with the POLDER-3/AERO-AC data (Waquet et al. 2013a; Waquet et al. 2013b; Peers et al. 2015). The purpose of this distinction is to better differentiate the various physical-chemical mechanisms involved between the emissions and transport of BBA (changes in the size and chemical composition of particles) impacting their optical properties. This differentiation also maximizes the number of

satellite observations available (see Figure 2).

We choose the RETRO (REanalysis of the TROposheric chemical composition, Schultz et al. 2007) global inventory for the anthropogenic emissions at a horizontal spatial resolution of 0.5x0.5 degrees and at a monthly temporal resolution over the 1960-2000 period. Biogenic emissions are calculated by WRF-Chem from surface, vegetation and weather data using MEGAN2.1 (Model of Emissions of Gases and Aerosols from Nature version 2.1, Guenther et al. 2012) at a horizontal spatial





resolution of 1x1 km² and at a monthly temporal resolution. Emissions of desert dust and sea salt particles are calculated online by WRF-Chem using the GOCART aerosol module from meteorological and surface parameters.

We use the APIFLAMEv1 (Analysis and Prediction of the Impact of Fires on Air quality ModEling, version 1.0) detailed French global inventory for BB emissions (Turquety et al. 2014), which is connected to a smoke plume rise model (Freitas et al. 2006, 2007; Grell et al. 2011). This time-dependent 1D cloud model explicitly simulates the rise of the plumes from the

source level in function of the convective transport mechanisms and the ambient weather and thermodynamic conditions calculated by WRF-Chem. APIFLAMEv1 calculates daily emissions of 49 chemical compounds produced by BB such as the BC, the OC or the carbon monoxide (CO, considered as a fire tracer) at a horizontal spatial resolution of 1x1 km². The methodology is based on burned areas and combustion temperatures from remote sensing data as well as emission factors specific to burnt vegetation from the Akagi et al. (2011) database. Comparison of APIFLAMEv1's CO emissions with those

from three other BB emission inventories often used in the scientific community, GFEDv3 (Global Fire Emissions Database, version 3.0 – Giglio and Randerson 2010; Werf et al. 2010), FINNv1 (Fire INventory from NCAR, version 1.0 – Wiedinmyer et al. 2011) and GFASv1 (Global Fire Assimilation System, version 1.0 – Kaiser et al. 2012), showed good spatial-temporal correlations of 0.9 (Turquety et al. 2014). These three global emission inventories also rely on MODIS observations for fire detection. The validation study of APIFLAMEv1 conducted by Turquety et al. (2014) showed uncertainties of a factor of 2 to

4 on the emitted amounts of carbonaceous species linked in particular to the wide variability of emission factors by vegetation type. However, these uncertainties are similar to those associated with other BB emission inventories (Wiedinmyer et al. 2011). It is worth noting that the global total emission amounts of carbonaceous aerosols differ by a factor of 3 to 4, ranging from 1.65 to 5.54 Tg for BC and from 13.76 to 51.93 Tg for OC (Pan et al. 2019). Figure 1 illustrates the monthly mass concentrations (in $g.m^{-2}$) of $PM_{10}$ emitted at surface by BB during the July-September 2008 period (from left to right) in

Africa from APIFLAMEv1 (Turquety et al. 2014) used in WRF-Chem V3.9.1.1.

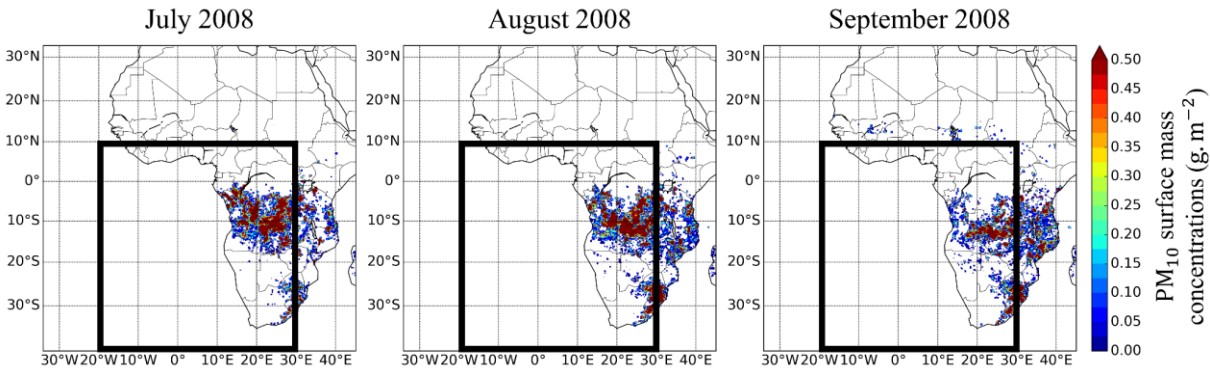

**Figure 1: Monthly mass concentrations of $PM_{10}$ (in $g.m^{-2}$) emitted at the surface by African biomass burning in July (left), August (middle) and September (right) 2008 from the APIFLAMEv1 biomass burning emission inventory (Turquety et al. 2014). The black frames of geographic coordinates (20°W-30°E, 39.4°S-10°N) represent the domain of study.**

Figure 1 indicates the location of wildfire sources with significant activity which are mainly in the southern half of Africa during the simulated period. During the dry season, which extends from May to October, the sources of wildfires gradually



move south and east, in agreement with previous studies in this region (Cahoon et al. 1992; Roberts et al. 2009). Figure 1 shows that the BB emission inventory of Turquety et al. (2014) reflects the expected variability in the detection of BBA sources in Africa.

## 2.2 A-Train aerosols and clouds satellite data

### 2.2.1 POLDER-3/PARASOL

The clear-sky Generalized Retrieval of Atmosphere and Surface Properties GRASP inversion algorithm allows the retrieval of the atmospheric properties of aerosols either from combined or separate observations from different passive and active remote sensing tools (satellite, ground, airborne) (Dubovik et al. 2011, 2014). Applied on POLDER-3's measurements of

linearly polarized radiance (490, 670 and 865 nm) and total radiance (443, 490, 565, 670, 865 and 1,020 nm), POLDER-3/GRASP allows to derive surface properties (e.g. albedo, reflectance), aerosol optical properties (e.g. total extinction optical depth, total absorption optical depth, fine mode optical depth, coarse mode optical depth, Angstrom exponent, single scattering albedo, refractive index), aerosol microphysical properties (e.g. particle size distribution, fine and coarse modes, spherical fraction of particles) and the average altitude of the aerosol layer. In this study, we will use the so-called "optimized" version

of POLDER-3/GRASP, which includes some approximations on radiative transfer calculations in order to obtain the best possible compromise between processing speed and accuracy of the results. This processing allowed the entire POLDER-3/PARASOL archive (March 2005-October 2013) to be analysed for the first time with this type of algorithm. Comparisons with coincident AERONET data have shown that the total AOD is retrieved within an uncertainty range of about ±15 % over land and ocean (Dubovik et al. 2011; Popp et al. 2016; Chen et al. 2018; 2019; 2020; Li et al. 2019). The maximum error on

the SSA was estimated to be about ±0.05 over land surfaces and in cases of high aerosol loads (Chen et al. 2020). In this study, we will use the total extinction optical depth at 565 nm and its spectral dependence (443, 490, 565, 670, 865 and 1,020 nm) derived from POLDER-3/GRASP in clear-sky conditions to assess the WRF-Chem model in terms of BB emission sources over land. The spectral retrievals of the single scattering albedo from 443 to 1,020 nm will be used to study the absorption properties and chemical composition of the emitted simulated aerosols.

The AEROsol Above Clouds POLDER-3/AERO-AC inversion algorithm retrieves the aerosol optical and microphysical properties above liquid water clouds by combining POLDER-3's multidirectional and multispectral measurements of polarized (670 and 865 nm) and total (490 and 865 nm) radiances (Waquet et al. 2013a,b; Peers et al. 2015). POLDER-3/AERO-AC retrieves the aerosol extinction optical depth ($ACAOD_{ext}$) and their single scattering albedo (ACSSA) at 490, 550, 670 and 865 nm and their Angstrom exponent ($ACAE_{670-865}$) simultaneously with the optical depth of the underlying cloud (COD) at a

spatial resolution of 6x6 km$^2$ and an almost daily temporal resolution. Retrievals are only provided in cases of optically thick ($COD > 3$) and homogeneous liquid water clouds, fractional cloud covers and cloud edges being eliminated. Scenes corresponding to cirrus clouds above liquid clouds are identified from coincident measurements acquired in thermal infrared by the MODIS instrument and are also discarded (Waquet et al. 2013a; Peers et al. 2015). This innovative inversion method





was the first to allow the retrieval of aerosol properties above cloudy scenes at a global scale (Waquet et al. 2013b). Sensitivity
studies for POLDER-3/AERO-AC indicate maximum relative errors of ±20 % on the extinction optical depth and ±0.05 on
the single scattering albedo of BB particles transported above liquid clouds (Peers et al. 2015). This inversion technique has
been compared with retrievals from the CALIOP lidar (Deaconu et al. 2017). The analysis of the extinction optical depth
retrieved by the CALIOP "depolarization" product (Hu et al. 2009) showed deviations of less than 20 % compared to
POLDER-3/AERO-AC on the ACAOD$_{ext}$ for the fine particles of BB. Differences between POLDER-3/AERO-AC and
CALIOP are increasing for mineral dust and especially for a complex mixture of aerosols (e.g., BB and urban pollution). In
this study, we will use the aerosol extinction optical depth at 550 nm and its spectral dependence (490, 550, 670 and 865 nm)
to assess the WRF-Chem model in terms of BBA content transported above marine stratocumulus over the SEAO. Spectral
retrievals of the single scattering albedo from 490 to 865 nm above clouds will be used to study the absorption properties and
chemical composition of simulated aerosols off the coasts of Southern Africa.

The aerosol parameters simulated with WRF-Chem have been spatially and temporally collocated with the remote sensing
parameters to be quantitatively comparable. First, the POLDER-3/GRASP and POLDER-3/AERO-AC data initially at a
horizontal spatial resolution of 6x6 km² are aggregated onto the model's grid (30x30 km²). Then, the POLDER-3's clear-sky
and cloudy sky masks were applied to the data simulated with WRF-Chem. Finally, the model meteorological-optical-chemical
parameters are averaged between 1 p.m. and 2 p.m. to be consistent with the PARASOL satellite transit time (about 1:30 p.m.
over the SEA region). Thus, each pixel simulated with WRF-Chem can be compared to a coincident aggregated pixel retrieved
by the POLDER-3 space sensor in clear-sky (GRASP) and above clouds (AERO-AC). Finally, the aerosol optical properties
simulated with WRF-Chem at four wavelengths (300, 400, 600 and 1,000 nm) in the shortwave radiation are recalculated at
the wavelengths of the POLDER-3 aerosol retrievals. The single scattering albedo (SSA) is linearly interpolated. The aerosol
optical depth (AOD) is interpolated using the Angström power law according to the following relationship:

$$AOD(\lambda_{interpol}) = AOD(\lambda_1)\left(\frac{\lambda_{interpol}}{\lambda_1}\right)^{-\frac{\ln\left(\frac{AOD(\lambda_1)}{AOD(\lambda_2)}\right)}{\ln\left(\frac{\lambda_2}{\lambda_1}\right)}} \tag{1}$$

$\lambda_{interpol}$ is the wavelength to be interpolated between $\lambda_1 = 400$ nm and $\lambda_2 = 600$ nm if 400 nm $< \lambda_{interpol} < 600$ nm or
between $\lambda_1 = 600$ nm and $\lambda_2 = 1,000$ nm if 600 nm $< \lambda_{interpol} < 1,000$ nm.

Figure 2 illustrates the number of observation days in clear (left) and cloudy (right) skies available from the POLDER-3 space
sensor during the first half of July 2008.





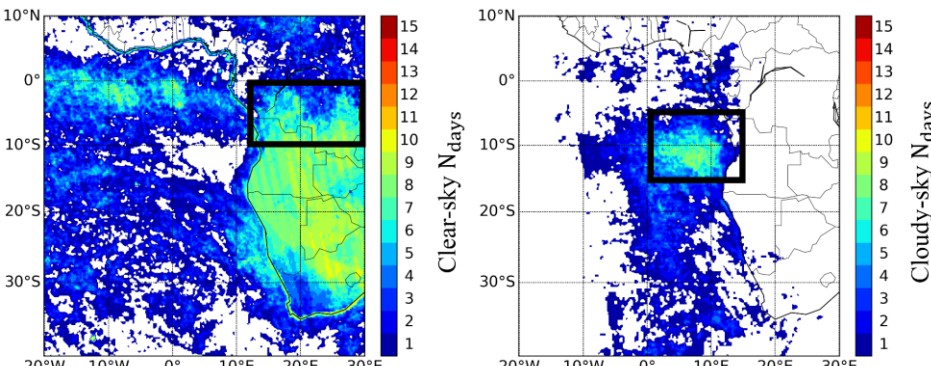

**Figure 2: Number of observation days ($N_{days}$) in clear atmosphere (left) and cloudy atmosphere (right) available from the POLDER-3 space sensor during the 01-15 July 2008 period. The black frames indicate the two areas of study of the biomass burning aerosol optical properties in clear-sky over land (12°E-30°E, 10°S-0°S, left) and above clouds over the Southeastern Atlantic Ocean (0°E-15°E, 15°S-5°S, right).**

Figure 2 indicates the state of the sky (clear or cloudy) over the SEA region during the first half of July 2008. This figure

shows the semi-permanent layer of marine stratocumulus inside the area of geographic coordinates (0°E-15°E, 15°S-5°S) characterized by the absence of satellite data in clear-sky (left) and by the largest number of satellite data in cloudy sky (right). Filters were applied to the POLDER-3 satellite observations to keep the signal associated with the lowest uncertainties on the retrieved aerosol optical properties. In clear-sky conditions, POLDER-3/GRASP aerosol optical depth and single scattering albedo were filtered according to the criterion $AOD_{obs}(443\ nm) \geq 0.4$ (Dubovik et al. 2011, 2014). Above cloudy scenes, we

used the POLDER-3 "AERO-AC-quality-assured-absorption-6km" product, which uses criteria to ensure the quality of the retrieval of the aerosol absorption above clouds (Waquet et al. 2020). These criteria restrict the retrievals of aerosol absorption above clouds to scenes with significant aerosol loading above clouds and optically thick clouds for which the sensitivity of the retrieval absorption method is maximal (Peers et al. 2015, 2016).

### 2.2.2 Ancillary data from CALIOP/CALIPSO

The operational inversion method applied to the CALIOP lidar level 2 data mainly retrieves the AOD in clear-sky and above clouds, the Angstrom Exponent (AE) above clouds and the vertical profiles of the aerosol backscatter and extinction coefficients from the backscattering signal measured at 532 and 1064 nm and the depolarization ratio (Winker et al. 2009). The altitudes of the base and the top of the clouds and the aerosols are also provided. In this study, we will use the vertical profiles of the aerosol extinction coefficient at 532 nm to evaluate that simulated with WRF-Chem at 550 nm as well as the

altitudes of the aerosol and cloud layers provided by CALIOP to assess their vertical distributions simulated with WRF-Chem.

### 2.3 Methodology

Our approach is summarized in Figure 3, with the main aerosol parameters investigated in the WRF-Chem numerical experiments reported in Table 2.



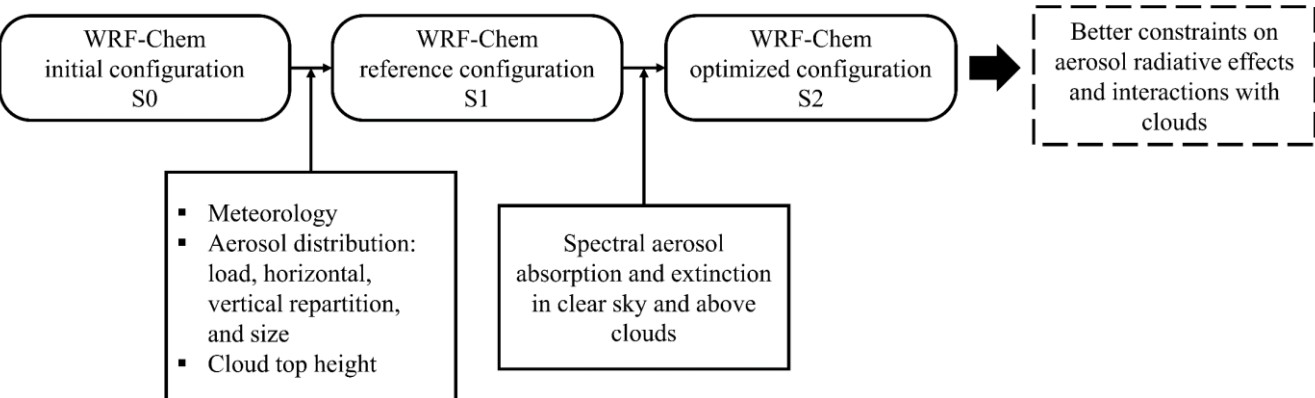

**Figure 3: Schematic diagram of the general approach applied to constrain WRF-Chem simulations of aerosols using A-Train satellite coincident retrievals of the distributions of aerosols and clouds.**

As a first step, we use the ECMWF reanalysis and the aerosol and cloud satellite observations from the A-Train to test the performance of the model in its initial configuration (S0, in Figure 3) in terms of simulations of meteorology, aerosol horizontal and vertical distribution, including their load. The simulated intensity of the BBA emissions over the South African sources is

evaluated using the AOD from POLDER-3/GRASP. Potential biases are then corrected by adjusting the emission inventories. Desert dust emissions from North African sources are also considered as they may significantly contribute to the total aerosol load over our studied area. Table 2 provides the uncertainties associated with the emissions of these two aerosol types (Turquety et al. 2014; Flaounas et al. 2016) along with the correction factors tested in the WRF-Chem simulations.

The injection height of BBA at the source level is evaluated using the aerosol vertical distribution provided by CALIOP. It

should be noted that the parameterization used in WRF-Chem (smoke plume rise model, Freitas et al. 2006, 2007; Grell et al. 2011) allow aerosols to be injected high enough into the atmosphere to simulate the transport of the BBA plumes above clouds over the SEAO. In addition, the simulation of the cloud top height in WRF-Chem is investigated, as this has a direct impact on the ability of the model to compute realistic above-cloud aerosol loads.

Finally, we evaluate the aerosol size distributions simulated with WRF-Chem, because of their known influence on the

calculation of the SSA. This analysis will help to separate the impact of the aerosol size from the aerosol chemical composition when comparing the simulated and the satellite derived SSA.

Following the evaluation of the model's performance, the possible biases are corrected by adjusting the emissions at the source level within their reported uncertainties. The reference configuration of WRF-Chem (S1, in Figure 3) corresponds to the configuration using the corrected emissions.

The second step of our approach consists in simultaneously constraining the load and the chemical composition of the BBA simulated with WRF-Chem using the spectral extinction (clear-sky and above-cloud AOD) and the spectral absorption (clear-sky and above-cloud SSA) from POLDER-3, in order to set up an optimized configuration of the model (S2, in Figure 3). Simulations are performed with WRF-Chem in its reference configuration using different amounts of BC, OC and BrOC at





the source level. The range of BC/OC and BrOC/OC ratios used in this experiment has been selected according to uncertainties

from the literature (Werf et al. 2010; Akagi et al. 2011) and are reported in Table 2. The set of BC, OC and BrOC amounts that reproduces the satellite observations the best corresponds to the optimized configuration (S2, in Figure 3). These simulation experiments also take into account complex refractive indices of pure species for BC and BrOC (see Table 2). We have tested two values of the BC refractive index. First, the one recommended by Bond and Bergstrom (2006), $m_{BC,550} = 1.95 - 0.79i$, which is commonly used as input data into climate models and implemented by default in WRF-Chem. Secondly,

we have tested the BC refractive index of Williams et al. (2007), $m_{BC,550} = 1.75 - 1.03i$, which is more absorbing and could be more realistic than that of Bond and Bergstrom (2006) since it underestimates by only 4 % (30 % for that of Bond and Bergstrom, 2006) the mass absorption cross section reference value ($MAC_{BC,fresh} = 7.5 \pm 1.2 \, m^2.g^{-1}$) for freshly emitted BC, i.e., unaged and uncoated (Bond and Bergstrom 2006; Liu et al. 2020).

The spectral BrOC refractive index provided by Hoffer et al. (2016, 2017), i.e., $m_{BrOC} = 1.86 - 0.25i$ at 550 nm, has been

used in our numerical experiments with three values of absorbing fractions of OC (0, 0.025 and 0.05) based on preliminary sensitivity tests. Hoffer et al. (2016, 2017) are the first, to our knowledge, to propose a direct experimental measurement of the absorption of BrOC particles from the ultraviolet to the near-infrared using an aethalometer with seven spectral bands (370-950 nm).

| | | Range of uncertainties reported in literature | Tested values in WRF-Chem simulations | |
|---|---|---|---|---|
| Emission inventories (coefficients applied to emissions) | Desert dust | $0.25 - 1^a$ | 0.5 | S1 |
| | Biomass burning | $2 - 4^b$ | 1.5 – 2 | S1 |
| Composition of biomass burning aerosols | BC/OC | $0.06 - 0.33^c$ | 0.06 – 0.25 | S2 |
| | BrOC/OC | $0 - 0.8^d$ | 0, 0.025, 0.05 | S2 |
| Imaginary part of refractive index 550 nm | BC | $0.45 - 1.1^e$ | $0.79 - 1.03^f$ | S2 |
| | BrOC | $0.003 - 0.27^g$ | $0.25^h$ | S2 |

**Table 2: Main parameters investigated, with range of values tested in simulations experiments performed with WRF-Chem,**
**considering the range of uncertainties reported in the literature. As indicated in Figure 3, S1 corresponds to the WRF-Chem reference simulation, S2 to the WRF-Chem optimized configuration. BC, OC and BrOC represent the black carbon, organic carbon and brown carbon in biomass burning aerosols emitted by WRF-Chem. [a]Flaounas et al. (2016); [b]Turquety et al. (2014); [c]Werf et al. (2010), Akagi et al. (2011); [d]Feng et al. (2013); [e]Liu et al. (2018); [f]Bond and Bergstrom (2006), Williams et al.( 2007); [g]Alexander et al. (2008), Sumlin et al. (2017); [h]Hoffer et al. (2016, 2017).**

It should be noted that other parameterizations in the model governing the life cycle of aerosols, such as aging, mixing, deposition, or entrainment, have not been modified or tested but they may also have an impact on the AOD and the SSA.



# 3 Results and discussion

## 3.1 Meteorology in the WRF-Chem initial configuration (S0)

Figure 4, Figure 5 and Figure 6 show the spatial distributions of monthly averaged wind speed (in $m.s^{-1}$) and direction,
specific humidity (in $g.kg^{-1}$) and temperature (in °C) respectively. The parameters at 850 hPa (left), 700 hPa (middle) and
500 hPa (right) for July 2008 simulated with WRF-Chem in its initial configuration (bottom) are compared to the ECMWF
reanalysis (top).

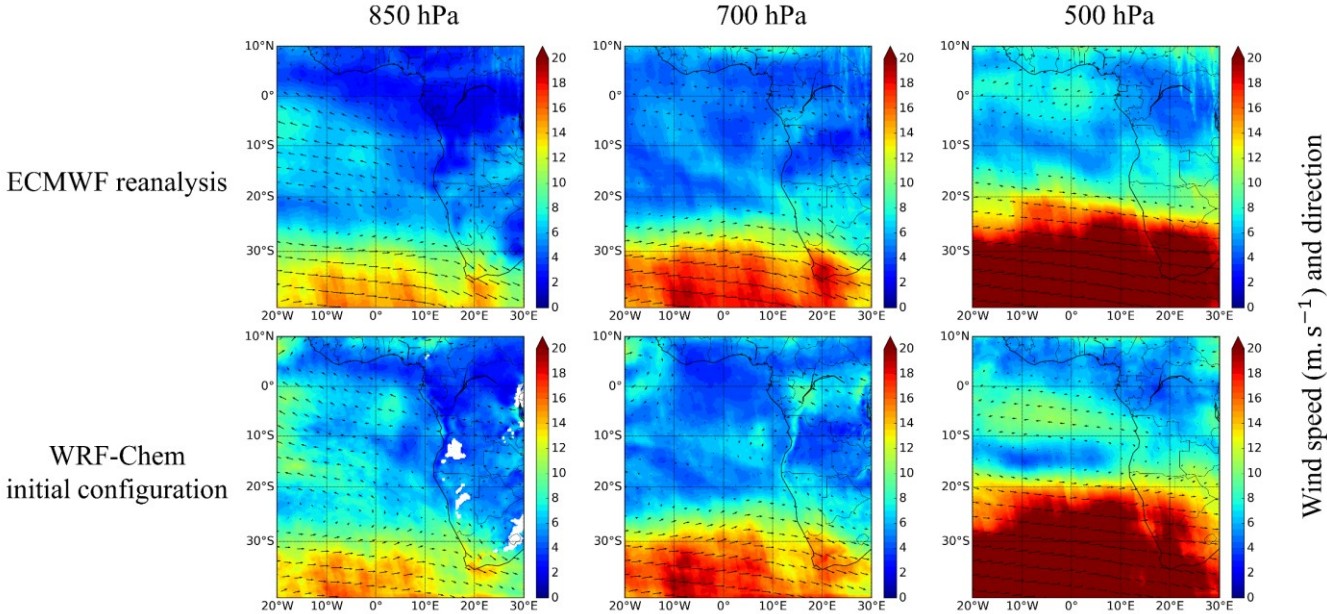

**Figure 4: Monthly averaged wind speed ($m.s^{-1}$) and direction at 850 hPa (left), 700 hPa (middle) and 500 hPa (right) from ECMWF**
**reanalysis (top) and simulated with the WRF-Chem initial configuration (bottom) for July 2008.**





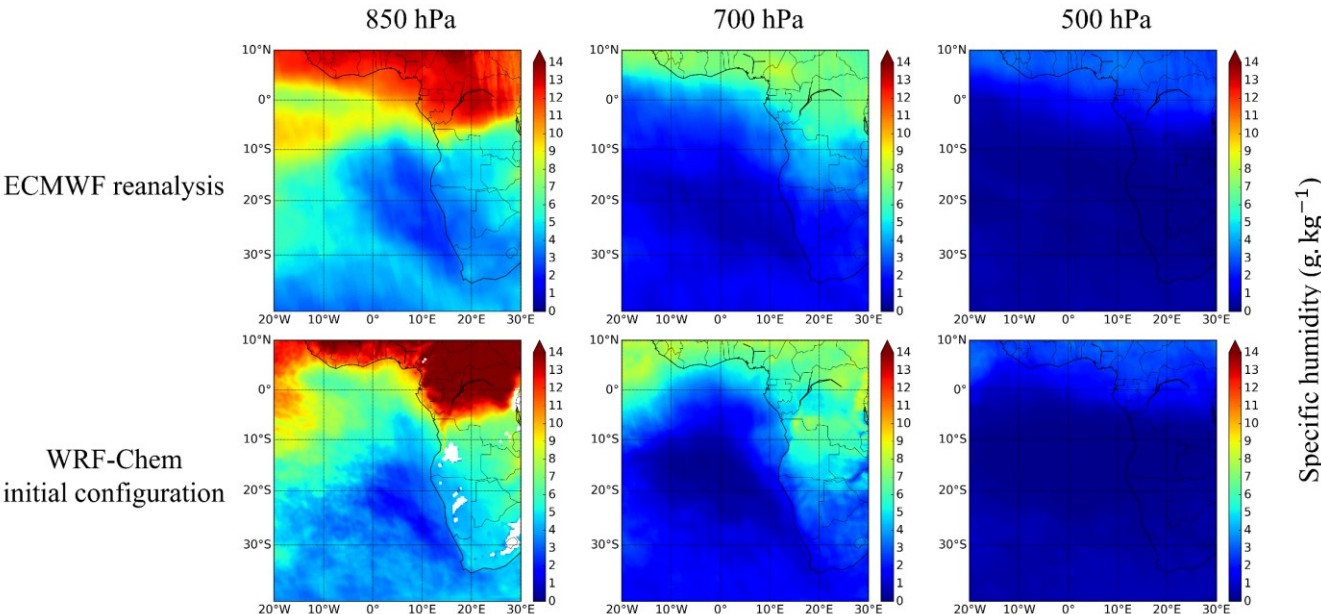

**Figure 5: Same as Figure 4 but for specific humidity (in g. kg$^{-1}$).**

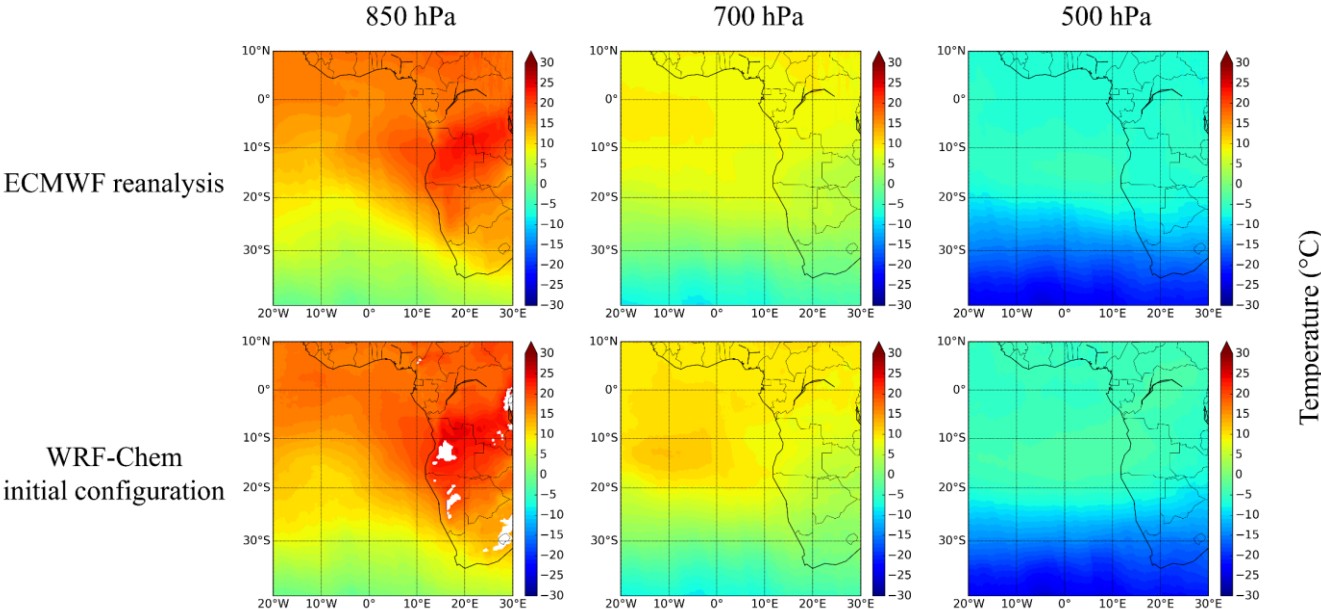

**Figure 6: Same as Figure 4 but for temperature (in °C).**

Figure 4, Figure 5 and Figure 6 show that the WRF-Chem initial configuration correctly simulates the general pattern of wind

speed, wind direction, humidity and temperature both in terms of magnitude and location at the three pressure levels, with only

local minor differences. North of the 20$^{th}$ parallel south, the mean simulated wind blows from east-south-east to west-north-



west due to the presence of anticyclonic conditions (Cahoon et al. 1992; Garstang et al. 1996; Swap et al. 1996), which favours a westward transport over the SEAO of BBA plumes emitted from the Southern African continent. The WRF-Chem initial

configuration slightly overestimates the magnitude of the wind speed with positive mean biases of only $1.18 \, \mathrm{m.s^{-1}}$ at 850 hPa, $0.68 \, \mathrm{m.s^{-1}}$ at 700 hPa, and $0.60 \, \mathrm{m.s^{-1}}$ at 500 hPa, and with low root mean square errors ($< 1.9 \, \mathrm{m.s^{-1}}$). The WRF-Chem initial configuration slightly underestimates the magnitude of the specific humidity at 850 hPa and 500 hPa with a negative mean bias of only $0.19 \, \mathrm{g.kg^{-1}}$ for both pressure levels and low root mean square errors ($\leq 1.1 \, \mathrm{g.kg^{-1}}$), which could impact the column-integrated/above-cloud AOD. Finally, the WRF-Chem initial configuration slightly overestimates the

magnitude of temperature for the three pressure levels with low positive mean biases of 0.93 °C at 850 hPa, 0.94 °C at 700 hPa and 1.17 °C at 500 hPa.

| | 850 hPa | | | | 700 hPa | | | | 500 hPa | | | |
|---|---|---|---|---|---|---|---|---|---|---|---|---|
| | WSPD ($\mathrm{m.s^{-1}}$) | WDIR (°) | Q ($\mathrm{g.kg^{-1}}$) | T (°C) | WSPD ($\mathrm{m.s^{-1}}$) | WDIR (°) | Q ($\mathrm{g.kg^{-1}}$) | T (°C) | WSPD ($\mathrm{m.s^{-1}}$) | WDIR (°) | Q ($\mathrm{g.kg^{-1}}$) | T (°C) |
| N | 31884 | 31884 | 31884 | 31884 | 31968 | 31968 | 31968 | 31968 | 31968 | 31968 | 31968 | 31968 |
| ME | 1.18 | -6.92 | -0.19 | 0.93 | 0.68 | 7.90 | 0.20 | 0.94 | 0.60 | -5.43 | -0.19 | 1.17 |
| MAE | 1.32 | 28.37 | 0.82 | 1.02 | 1.14 | 52.05 | 0.58 | 1.00 | 1.48 | 16.33 | 0.31 | 1.23 |
| RMSE | 1.70 | 1.01 | 1.10 | 1.21 | 1.57 | 1.60 | 0.83 | 1.29 | 1.82 | 0.75 | 0.41 | 1.41 |
| CORR | 0.93 | 0.96 | 0.95 | 0.99 | 0.95 | 0.90 | 0.94 | 0.99 | 0.97 | 0.98 | 0.93 | 0.99 |

**Table 3: Meteorology performance statistics of WRF-Chem initial configuration compared to ECMWF reanalyses. WSPD, WDIR, Qs and T represent the wind speed, the wind direction, the specific humidity and the temperature respectively. N, ME, MAE, RMSE and CORR represent the number of coincident data pairs, the mean error, the mean absolute error, the root mean square error and**
**the spatial Pearson correlation coefficient respectively. The units of the statistical parameters are the same as meteorological variables.**

Table 3 summarizes the statistics of the comparisons between the simulated and ECMWF wind speed (WSPD, in $\mathrm{m.s^{-1}}$), wind direction (WDIR, in °), specific humidity (Q, in $\mathrm{g.kg^{-1}}$) and temperature (T, in °C) at 850 hPa, 700 hPa and 500 hPa on average over the SEA region for July 2008. The statistical results shown in Table 3 confirm the overall good agreement between

the WRF-Chem initial configuration and the ECMWF reanalysis with spatial correlation coefficients between 0.90 and 0.99. In summary, this evaluation strongly suggests that the WRF-Chem initial configuration is able to correctly reproduce the local meteorology for the domain and the period considered in this study.

## 3.2 Aerosol distribution and size in the WRF-Chem reference configuration (S1)

### 3.2.1 Strength of aerosol emissions

The column integrated AOD that reflects the vertically integrated aerosol concentration is used here to assess the strength of the simulated aerosol emission sources. Figure 7 illustrates the spatial distribution of monthly averaged AOD at 565 nm in





clear-sky retrieved by POLDER-3/GRASP (left) and simulated with the initial configuration (middle) and with the reference configuration (right) of WRF-Chem for July 2008.

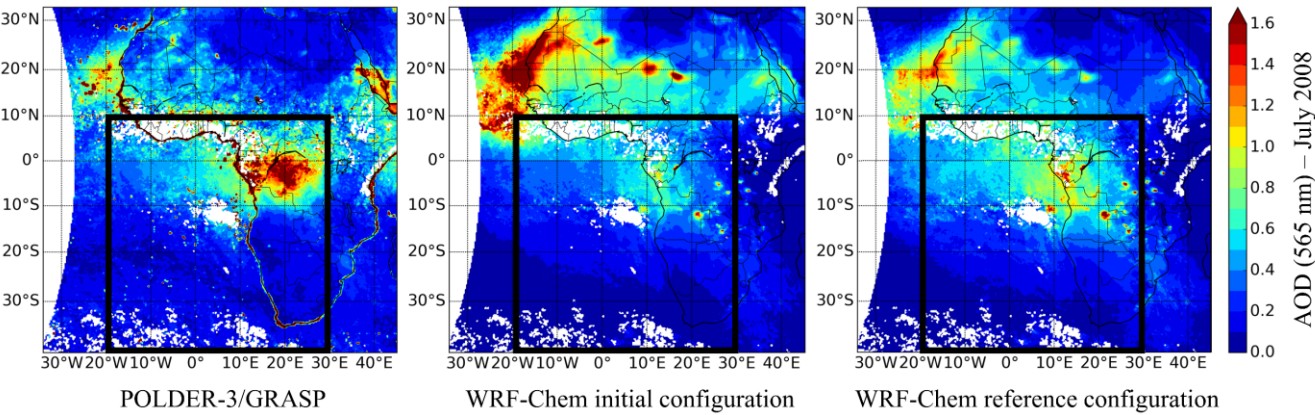

**Figure 7: Monthly averaged aerosol optical depth (AOD) at 565 nm in clear-sky retrieved by POLDER-3/GRASP (left) and simulated with the WRF-Chem initial configuration (middle) and with the WRF-Chem reference configuration (right) for July 2008. The black frames of geographic coordinates (20°W-30°E, 39.9°S-10°N) represent the domain of study.**

Figure 7 shows that the WRF-Chem initial configuration (mean $AOD_{mod,565}$ = 0.49) significantly underestimates, by a factor of 2 on average, the AOD retrieved by POLDER-3/GRASP (mean $AOD_{obs,565}$ = 1.10) at 565 nm in clear-sky over the BB emission sources (12°E-30°E, 10°S-0°S) for July 2008. This bias may could be due to an underestimation of the fire activity in the BB emission inventory. The assessment of APIFLAMEv1 by Turquety et al. (2014) indicates possible biases of a factor of 2 to 4 on all amounts of emitted gas and particulate species by BB. Figure 7 also shows that, over the northern half of Africa, in the Sahara and Sahel region, the average values of AOD are highly overestimated in the WRF-Chem initial configuration (mean $AOD_{mod,565}$ ≥ 1.5) compared to the POLDER-3/GRASP retrievals (mean $AOD_{obs,565}$ ≃ 1.0). The GOCART AFWA desert dust emission module used in WRF-Chem (Jones et al. 2010, 2012), based on the Marticorena and Bergametti (1995) scheme, seems to raise too much mineral dusts over the Sahara/Sahel area.

Therefore we have chosen to correct the APIFLAMEv1 BB emission inventory with a moderate multiplicative factor of 1.5 on all the gas and particulate species, in agreement with the uncertainties provided by Turquety et al. (2014). This scaling is a common practice in climate modelling studies for BBA, with emission factors generally higher than 1 (up to 6, Reddington et al. 2016) to better reproduce the observed satellite AODs (Johnson et al. 2016). According to the recommendations of Flaounas et al. (2016) who made an assessment of atmospheric dust modelling performance by WRF-Chem (version 3.6) against MODIS observations on arid and semi-arid regions around the Mediterranean, we have applied the adjustment coefficient of 0.5 on the desert dust emission surface fluxes in the GOCART AFWA scheme. This corrected version of WRF-Chem, in terms of BB and desert dust emissions, is the so-called reference configuration (S1).

Figure 7 shows that the WRF-Chem reference configuration better reproduces the magnitude of both BB and desert dust emission sources over land and their transport above ocean compared to WRF-Chem in its initial configuration, even if some biases still remain.



### 3.2.2 Aerosols and clouds vertical distribution

Besides simulated the aerosol loads, the major factors that may influence the WRF-Chem ACAOD are the injection height of
BBA, the cloud top height, and the vertical distribution of both aerosols and clouds.

The injection height of BBA conditions their transport and their potential interactions with cloud layers (Hansen et al. 1997; Podgorny and Ramanathan, 2001; Rosenfeld et al. 2014; Lee et al. 2016). BBA simulated with the WRF-Chem reference configuration are injected into the atmosphere up to an altitude of about 6 km, consistently with CALIOP observations analysed in this part of the world (Koffi et al. 2012, 2016), which allows them to be transported over a long distance. One of WRF-
Chem's strengths for the simulation of BBA is that their injection height is dynamically calculated according to the thermodynamic conditions of fires, whereas it is generally prescribed in climate models (Freitas et al. 2006, 2007; Grell et al. 2011).

The cloud top height from satellite is based on the POLDER oxygen pressure method (Vanbauce et al. 2003; Ferlay et al. 2010; Deaconu et al. 2019). On the other hand, the estimation of the cloud top height is not directly provided by WRF-Chem. As a
first approach, we have only selected clouds with a vertical monolayer structure in the model, i.e., for which the simulated vertical profile of the cloud liquid water content (QCLOUD) has one maximum. We have qualified these clouds as homogeneous (in the vertical sense) in the following. Clouds with a complex vertical structure have been qualified as heterogeneous. For these last situations, the simulated QCLOUD vertical profile has several maximums and the top of the stratocumulus cannot, therefore, be easily defined. The main advantage of this technique is to guarantee the selection of
uniform monolayer cloud situations, consistent with data obtained from the POLDER-3/AERO-AC inversion algorithm (Waquet et al. 2013a,b; Peers et al. 2015). Situations with cirrus clouds are rejected as in the POLDER-3/AERO-AC product. The cloud top height is then determined by scanning the cloud profile from the bottom to the top and stopping at a threshold value of liquid water content (LWC). A statistical study was carried out to determine the optimal value of this threshold. We have considered threshold values between 50 % and 100 % corresponding to a fraction of the vertically integrated QCLOUD
in the WRF-Chem reference configuration. This analysis showed that, statistically, the closest results to the POLDER-3 retrievals were obtain with a threshold of 95 %. Below this value, the cloud top heights simulated with the WRF-Chem reference configuration shift towards the lowest value classes, enhancing the underestimation of the simulated values compared to the POLDER-3 retrievals. For a threshold of 100 %, the number of cases for the highest classes increases, enhancing the overestimation of the values simulated with the WRF-Chem reference configuration in comparison with those retrieved by
POLDER-3. Figure 8 illustrates the spatial distribution of monthly averaged cloud top heights retrieved by POLDER-3 (left) and simulated with the WRF-Chem reference configuration with the threshold of 95 % (right) for July 2008.

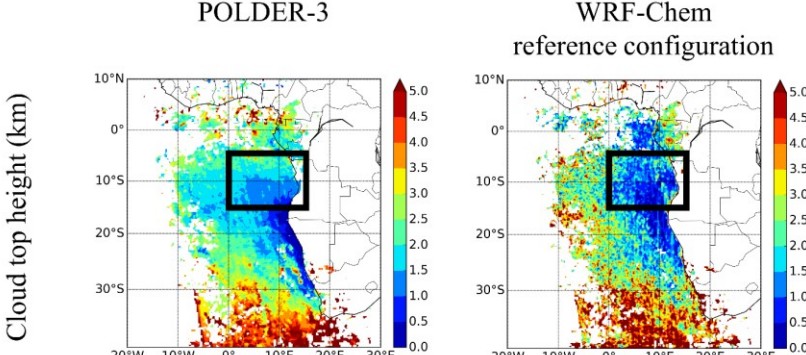

**Figure 8: Monthly averaged cloud top heights retrieved by POLDER-3 (left) and simulated with the WRF-Chem reference configuration with the threshold of 95 % (right) for July 2008. The black frames of geographic coordinates (0°E-15°E, 15°S-5°S)**
**represent the study area.**

Figure 8 shows that along Angola and Namibia, the WRF-Chem reference configuration well simulates the low cloud area typically observed in these regions (Deaconu et al. 2019). In the studied area (black frame), Figure 8 shows that the WRF-Chem reference configuration slightly underestimates the mean altitudes of the cloud top compared to the POLDER retrievals. South of the studied area as well as north over land (Congo, Gabon), cloud convection is fairly well reproduced with the WRF-
Chem reference configuration with a mean cloud top height above 2 km. On the opposite, a major difference is observed to the west of the studied area, with the WRF-Chem reference configuration simulating cumulus clouds with a mean cloud top height greater than 2.5 km, unlike POLDER-3 that retrieves low clouds with a mean cloud top height around 1.5-2.0 km. The difference between the model and the POLDER-3 retrievals could come from the cloud parameterization (Lin et al. 1983) used in the WRF-Chem reference configuration or this could be due to the fact that the aerosol feedbacks on clouds are not taking
into account in the model.

Figure 9 illustrates the vertical profiles of the aerosol extinction coefficient ($\sigma_{ext}$, in km$^{-1}$) retrieved by CALIOP at 532 nm (orange dotted lines) and simulated with the WRF-Chem reference configuration at 550 nm (red solid lines) for July 12, 2008 (left) and July 28, 2008 (right). These two specific days were selected due to dense BBA plumes transported above marine clouds over the SEAO. The extinction of aerosols retrieved by CALIOP above clouds has been corrected with POLDER-
3/AERO-AC according to Deaconu et al. (2019). Deaconu et al. (2017) have shown that CALIOP underestimates the aerosol load above clouds by a factor of 2 to 4 in comparison with the POLDER-3/AERO-AC retrievals and other advanced CALIOP products such as the so-called depolarization method (Hu et al. 2009). Vertical profiles of the simulated aerosol extinction coefficients are averaged under the CALIOP track aggregated over the area of geographic coordinates (0°E-15°E, 15°S-5°S). This area is characterized by the presence of a semi-permanent layer of marine stratocumulus (see Figure 2). The mean cloud
top heights retrieved by CALIOP (green dotted lines) and simulated with the WRF-Chem reference configuration (blue solid lines) are also shown in Figure 9.





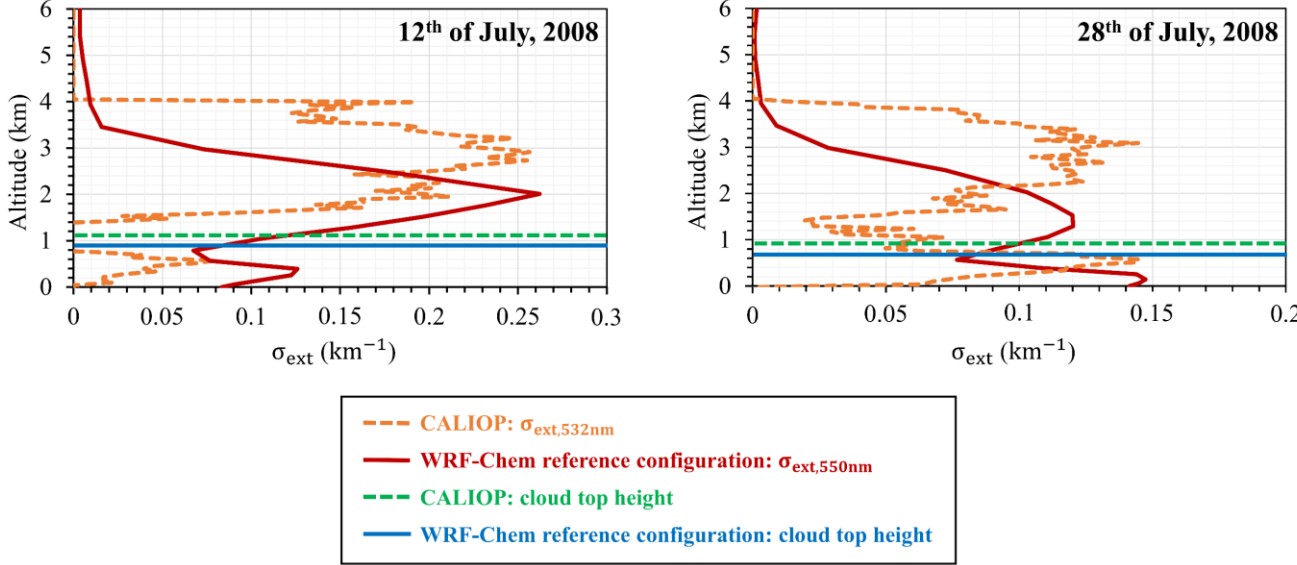

**Figure 9: Mean vertical profiles of the aerosol extinction coefficients retrieved by CALIOP at 532 nm (orange dotted lines) and simulated with the WRF-Chem reference configuration at 550 nm (red solid lines) under the CALIOP track inside the study area of**

**geographic coordinates (0°E-15°E, 15°S-5°S) for July 12, 2008 (left) and July 28, 2008 (right). Mean cloud top heights retrieved by CALIOP (green dotted lines) and simulated with the WRF-Chem reference configuration (blue solid lines) are also incorporated into the graphics. CALIOP data, initially at a horizontal spatial resolution of 5x5 km$^2$, were aggregated onto the WRF-Chem mesh (30x30 km$^2$) and only the coincident pixels were selected for the comparison study.**

In Figure 9, similarities are observed between the vertical profiles from CALIOP and from the WRF-Chem reference

configuration with two distinct aerosol layers and the presence of a low cloud layer for both case studies. However, mean altitudes of the aerosol and the cloud layers differ slightly and appear to be systematically underestimated in the WRF-Chem reference configuration compared to the CALIOP retrievals. For July 12, 2008, the first aerosol layer located between 0 and 0.8 km above sea level retrieved by CALIOP and simulated with the WRF-Chem reference configuration could correspond to a mixture of aerosols including sea salts. The second one, which is located between 1.4 and 4.0 in the CALIOP observations

and between 0.8 and 3.4 km in the simulation from the WRF-Chem reference configuration, corresponds to BBA plumes usually observed in this region at this time of the year (Koffi et al. 2012, 2016; Zuidema et al. 2016; Formenti et al. 2019). Regarding clouds, the WRF-Chem reference configuration underestimates by about 200 m (300 m) the mean cloud top height on July 12, 2008 (28 July 2008) compared to the CALIOP retrievals. Even if the altitudes of the simulated BBA layers are too low in comparison with the CALIOP retrievals, most of the modelled aerosol extinction is located above clouds and therefore,

we can conclude that the WRF-Chem reference configuration reproduces well the aerosol load integrated above clouds.

### 3.2.3 Particle size

The SSA of aerosols, which accounts for their absorption properties, depends on both the particles size and their chemical composition (Abel et al. 2003; Laing et al. 2016). The aerosol size can be qualitatively estimated by the AE, the highest values corresponding to the finest particles (Ångström 1929). Figure 10 shows the evolution of the vertically integrated volume size

distribution of aerosols simulated with the WRF-Chem reference configuration during the transport of BBA plumes averaged

over July 2008.

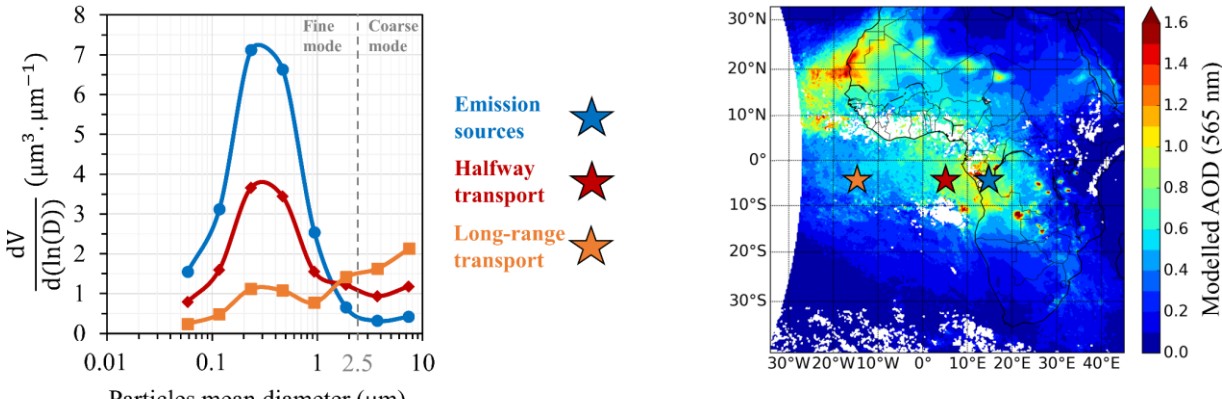

**Figure 10: Evolution of the vertically integrated volume size distribution of aerosols simulated with the WRF-Chem reference configuration for three areas representative of the emission sources and transport progress of the biomass burning aerosol plume**
**on average for July 2008. The blue star corresponds to the pixel of geographic coordinates (14.83°E, 4.00°S) and symbolizes the emission sources of biomass burning particles. The red star corresponds to the pixel of geographic coordinates (4.82°E, 4.00°S) and symbolizes the biomass burning aerosol plume transported halfway westward over the Southeastern Atlantic Ocean. The orange star corresponds to the pixel of geographic coordinates (14.93°W, 4.00°S) and symbolizes the long-range transport of the biomass burning aerosol plume.**

Figure 10 indicates that fine mode particles dominate the aerosol volume size distribution near BBA emission sources with a

mean particle diameter centred at 0.3 µm corresponding to the accumulation mode. During the aerosol transport, there is a

gradual appearance of larger particles (diameter > 1 µm), which may be linked to aging processes and also to a slightly more

pronounced influence of other larger particles, such as sea salts and desert dust aerosols, that may come across the BBA plumes

over the SEAO.

Figure 11 shows the spatial distributions of the AE retrieved by POLDER-3 (left) and simulated with the WRF-Chem reference

configuration (right) in clear-sky ($AE_{670-865}$, top) and above clouds ($ACAE_{670-865}$, bottom) on average for July 2008.



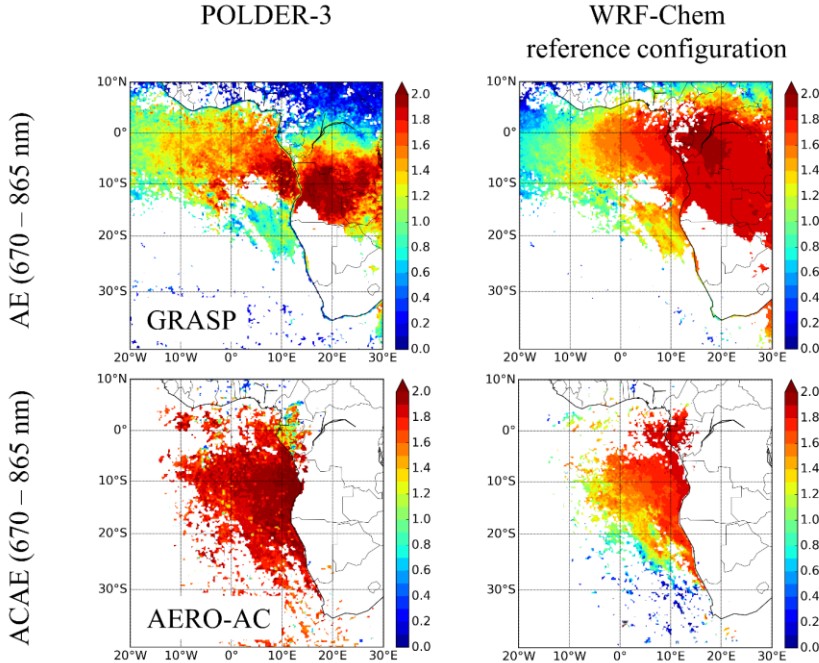

**Figure 11: Monthly averaged Angström exponent retrieved by POLDER-3 in clear-sky with GRASP (AE$_{670-865}$, top) and above clouds with AERO-AC (ACAE$_{670-865}$, bottom) (left) and simulated with the WRF-Chem reference configuration (right) for July**

**2008.**

Figure 11 shows a good agreement between the POLDER-3/GRASP retrievals (top, left) and the WRF-Chem reference simulation in clear-sky (top, right) with simulated AE$_{670-865}$ values greater than 1 over the SEA region and close to 2 over BB emission sources, these values being very comparable to those derived from POLDER-3. These values are typical of fine BB particles usually observed in this region (Dubovik et al. 2002). Above clouds over the SEAO, Figure 11 suggests a decrease

of the ACAE associated to the westward progression of the BBA plumes, particularly in the WRF-Chem reference configuration (bottom, right). Such an evolution could be related to a gradual growth of BB particles size and/or changes of the overall chemical composition of BBA plumes. Indeed, a decrease in the relative amount of fine BB particles and an increase of the proportion coarse mode particles (sea salts and desert dusts) that could occur during the transport of BBA plumes over the SEAO, as shown in Figure 10.

Overall, Figure 11 indicates that the particle sizes simulated with the WRF-Chem reference configuration are realistic, considering BBA plumes in clear-sky and above clouds, with only few minor differences compared to the POLDER-3 retrievals. Therefore, in the following, we assume that the potential difference between the simulated and the observed SSA is primarily due to the chemical composition.





### 3.3 Sensitivity analysis of aerosol absorption in the WRF-Chem reference configuration

Figure 12 illustrates the monthly averaged PM$_{2.5}$ chemical composition simulated with the WRF-Chem reference configuration, including organic aerosols (OA, in green), secondary inorganic aerosols (SIA, in grey), organic carbon (OC, in brown), black carbon (BC, in black), desert dust (DUST, in yellow) and sea salts (SEAS, in blue), and their relative proportions over the Southern African continent in clear-sky (spatial average over 12°E-30°E, 10°S-0°S, left) and above clouds over the SEAO (spatial average over 0°E-15°E, 15°S-5°S, right) for July 2008.

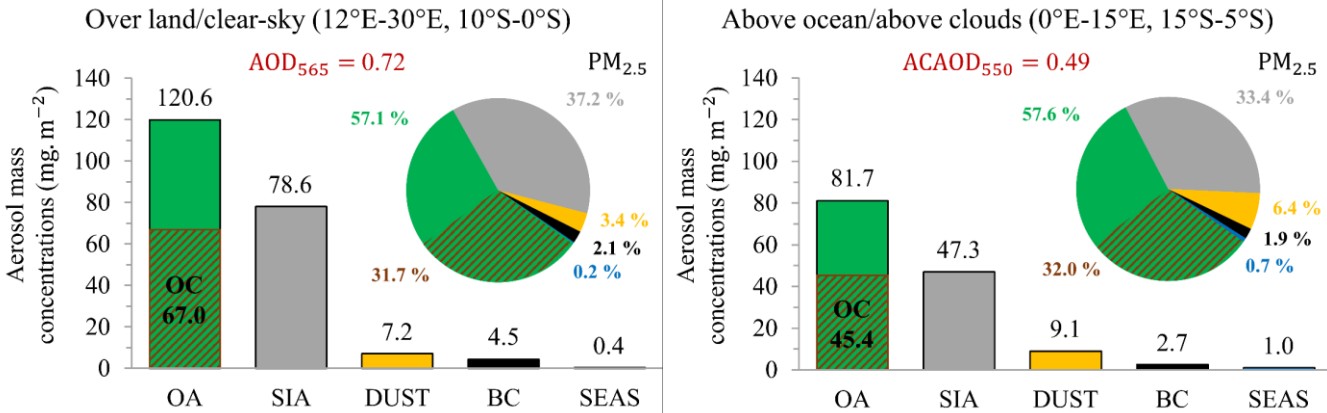

**Figure 12: Evolution of the PM$_{2.5}$ chemical composition simulated with the WRF-Chem reference configuration between the Southern African continent in clear-sky (12°E-30°E, 10°S-0°S, left) and the Southeastern Atlantic Ocean above clouds (0°E-15°E, 15°S-5°S, right). Mass concentrations (in mg. m$^{-2}$) of the main aerosol compounds of biomass burning, i.e. black carbon (BC, black), organic carbon (OC, brown hatching), organic aerosols (OA, green) and secondary inorganic aerosols (SIA, grey), as well as desert**
**dust (DUST, yellow) and sea salts (SEAS, blue), are vertically integrated over the atmospheric column on average for July 2008.**

In the region of BB emission sources (12°E-30°E, 10°S-0°S), organic aerosols (OA, green) are the majority species with a vertically integrated mass of 120.6 mg. m$^{-2}$ and representing 57.1 % of the total composition of PM$_{2.5}$. Secondary inorganic aerosols (SIA, grey), i.e., sulphates, nitrates and ammonium, are the second dominant species representing 37.2 % of the total composition of PM$_{2.5}$ with a vertically integrated mass of 78.6 mg. m$^{-2}$, followed by organic carbon (OC, brown hatching)

with a vertically integrated mass of 67.0 mg. m$^{-2}$ (relative proportion of 31.7 %). Vertically integrated mass concentration of black carbon (BC, black) is 4.5 mg. m$^{-2}$ (relative proportion of 2.1 %). During the transport of the BBA plumes above clouds over the SEAO (0°E-15°E, 15°S-5°S), mass concentrations of these BBA decrease due to the deposition processes. Figure 12 shows that the influence of desert dust and sea salt aerosols, although relatively low, increases during transport over the SEAO, from 7.2 mg. m$^{-2}$ to 9.1 mg. m$^{-2}$ for desert dusts and from 0.4 mg. m$^{-2}$ to 1.0 mg. m$^{-2}$ for sea salts.

The chemical compositions of aerosols simulated with the WRF-Chem reference configuration appear realistic in comparison with the recent observations obtained from DACCIWA (Dynamics-Aerosol-Chemistry-Cloud Interactions in West Africa, Flamant et al. 2017) in Southern West Africa (i.e. further north). Specifically, results obtained during the DACCIWA airborne campaigns showed that OA dominated by over 50 % the total mass of aerosols, that SIA contributed to about 34 % and that



the contribution of BC was under 15 % (Haslett et al. 2019). However, we can note that BC contributions simulated the WRF-
Chem reference configuration ($\leq$ 2.1 %) in our studied areas are smaller than those measured during DACCIWA.

Figure 13 illustrates the monthly averaged SSA in clear-sky (top) and above clouds (bottom) retrieved by the GRASP/AERO-AC inversion algorithms (left) and simulated with the WRF-Chem reference configuration (right) at 565/550 nm respectively for July 2008.

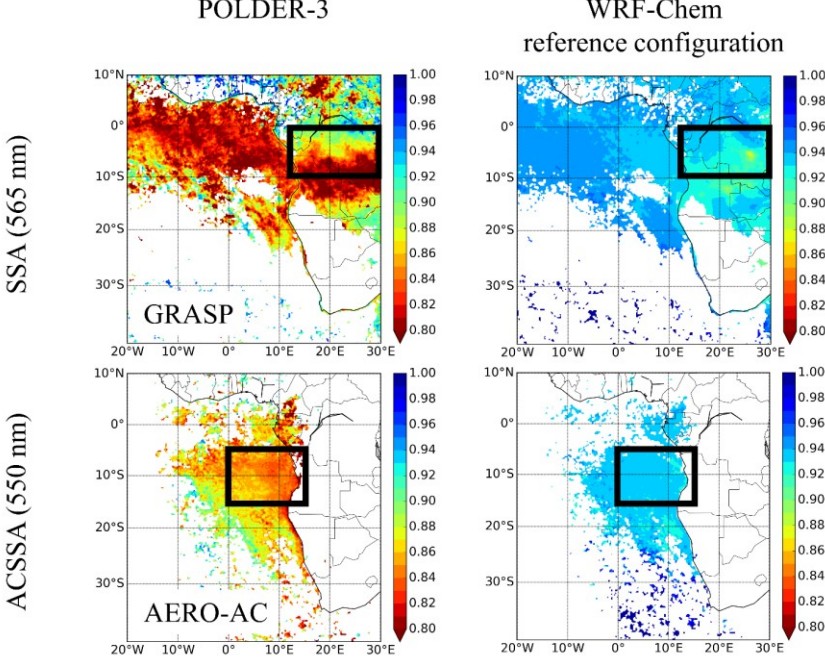

**Figure 13: Monthly averaged single scattering albedo of aerosols at 565 nm in clear-sky (SSA, top) and at 550 nm above clouds (ACSSA, bottom) retrieved by POLDER-3 (left) and simulated with the WRF-Chem reference configuration (right) for July 2008. The black frames of geographic coordinates (12°E-30°E, 10°S-0°S) and (0°E-15°E, 15°S-5°S) represent respectively the two study regions, in clear-sky (top) and above clouds (bottom).**

Figure 13 clearly shows that the aerosols simulated with the WRF-Chem reference configuration, in which only BC absorbs
solar radiation in the BBA plumes, are too scattering with a mean positive bias of about 0.08 between the simulated SSA and the one retrieved by POLDER-3, both in clear-sky and above clouds, for July 2008. In the continental area in clear-sky (12°E-30°E, 10°S-0°S), the mean SSA simulated with the WRF-Chem reference configuration is 0.93 at 565 nm while the SSA retrieved by POLDER-3/GRASP is 0.85 at 565 nm. In the oceanic area above clouds (0°E-15°E, 15°S-5°S), the mean SSA simulated with the WRF-Chem reference configuration is 0.93 at 550 nm, which is well above the value of 0.85 at 550 nm
retrieved by POLDER-3/AERO-AC. Furthermore, we can also observe a strong disagreement between the SSA from the WRF-Chem reference configuration and the SSA retrieved by POLDER-3/GRASP over the SEAO in clear atmosphere. On average, POLDER-3/GRASP retrieves a very low SSA of around 0.80 at 565 nm in this region that the WRF-Chem reference configuration fails to reproduce. To explain the large discrepancies between the WRF-Chem reference configuration and the POLDER-3 retrievals, four hypotheses can be made. (i) There is not enough BC in the BBA plumes simulated with the WRF-



Chem reference configuration. (ii) There is too much scattering organic aerosols in the simulated BBA plumes. (iii) The refractive index of BC used in the WRF-Chem reference configuration is not realistic enough. (iv) The presence of other absorbing species in the BBA plumes, such as the BrOC, needs to be considered in the WRF-Chem reference configuration. Thus, we have analysed the sensitivity of the SSA to the chemical composition of BB carbonaceous aerosols through a set of numerical simulations as a Look Up Table (reported Table 4).

| Adjustment factors | | | BC/OC | | |
|---|---|---|---|---|---|
| Source level | | | Surface | Vertically integrated | |
| BC | OC | BrOC[3] (%) | Emissions (12°E-30°E, 10°S-0°S) | Over land/Clear-sky (12°E-30°E, 10°S-0°S) | Above ocean/Above clouds (0°E-15°E, 15°S-5°S) |
| x1[1] | /1 | | 0.06 | 0.06 | 0.06 |
| | /5 | 0, 2.5, 5 | 0.14 | 0.15 | 0.11 |
| | /10 | | 0.17 | 0.19 | 0.12 |
| | /15 | | 0.18 | 0.20 | 0.13 |
| x2[1,2] | /1 | | 0.09 | 0.10 | 0.08 |
| | /2.5 | | 0.14 | 0.15 | 0.11 |
| | /5 | 0, 2.5, 5 | 0.17 | 0.19 | 0.12 |
| | /10 | | 0.19 | 0.22 | 0.13 |
| | /15 | | 0.20 | 0.23 | 0.13 |
| x2.5[2] | /1 | | 0.10 | 0.11 | 0.09 |
| | /2.5 | | 0.15 | 0.16 | 0.11 |
| | /5 | 0, 2.5, 5 | 0.18 | 0.20 | 0.13 |
| | /10 | | 0.20 | 0.22 | 0.13 |
| | /15 | | 0.21 | 0.24 | 0.14 |
| x3[2] | /1 | | 0.11 | 0.12 | 0.09 |
| | /2.5 | | 0.16 | 0.17 | 0.12 |
| | /5 | 0, 2.5, 5 | 0.19 | 0.21 | 0.13 |
| | /10 | | 0.20 | 0.23 | 0.13 |
| | /15 | | 0.21 | 0.24 | 0.14 |
| x3.5[1] | /1 | | 0.12 | 0.13 | 0.10 |
| | /5 | 0, 2.5, 5 | 0.19 | 0.21 | 0.13 |
| | /10 | | 0.20 | 0.23 | 0.14 |
| | /15 | | 0.21 | 0.24 | 0.14 |
| x4[1] | /1 | | 0.13 | 0.14 | 0.10 |
| | /5 | 0, 2.5, 5 | 0.19 | 0.22 | 0.13 |
| | /10 | | 0.21 | 0.24 | 0.14 |
| | /15 | | 0.21 | 0.25 | 0.14 |

[1]$m_{BC,550} = 1.95 - 0.79i$ (Bond and Bergstrom 2006)

[2]$m_{BC,550} = 1.75 - 1.03i$ (Williams et al. 2007)

[3]$m_{BrOC,550} = 1.86 - 0.25i$ (Hoffer et al. 2016, 2017)

**Table 4: Table of numerical experiments performed with the WRF-Chem reference configuration to test the impact of a range of organic carbon (OC), black carbon (BC) and brown carbon (BrOC) proportions on simulated biomass burning aerosols properties.**





We have considered adjustment factors up to 4 for BC and down to 1/15 for primary OC (POC) at the source level, consistently with the uncertainties in the APIFLAMEv1 BB emission inventory (Turquety et al. 2014). For example, the possible underestimation of BC emissions could be due to an underestimation of the BC emission factor for savannahs (0.37±0.20 g. kg⁻¹) in Akagi et al. (2011) used in APIFLAMEv1 compared to the more recent BC emission factor of 0.53±0.35 g. kg⁻¹ for savannahs in Andreae (2019). Although the 1/10 and 1/15 reduction factors of POC can be considered outside of the POC emission uncertainties (70 % of uncertainty in APIFLAMEv1), the corresponding simulations are used to show the impact of an increased of the aerosol absorption due to a strong increase of the BC fraction. Each couple of BC and OC adjustment factors is associated with a varying contribution of BrOC equals to 0 %, 2.5 % or 5 %. The BC/OC mass mixing ratio at the surface for all numerical experiments ranges from 0.06 to 0.21 and remains in the order of magnitude of those reported in the literature for African BB (see Table 2). These values are generally around 0.13-0.14 during the transport of the BBA plumes above clouds over the SEAO, consistently with Southern Africa airborne measurements (Formenti et al. 2003; Kirchstetter et al. 2003; Capes et al. 2008).

The selection process of the best WRF-Chem solution is described as follow. A score is computed for each WRF-chem numerical simulation. The score is equal to the number of times the result of the simulation is included within the range of the retrieval uncertainties. This score is computed over all the pixels for the two selected areas (with and without clouds) and over all available spectral aerosol parameters (i.e., AOD, SSA, 4 spectral bands for AERO-AC and 6 for GRASP).

As a result, with the BC refractive index of Bond and Bergstrom (2006), $m_{BC,550} = 1.95 - 0.79i$, we were unable to find a realistic solution of WRF-Chem within the uncertainties of APIFLAMEv1 (Turquety et al. 2014). With the BC refractive index of Williams et al. (2007), $m_{BC,550} = 1.75 - 1.03i$, an optimized configuration of WRF-Chem was obtained. It corresponds to the combination of the following carbonaceous aerosol mixture: BCx2, OC/2.5 with 2.5 % of BrOC. This WRF-Chem configuration is the one that statistically reproduces the spectral AOD and SSA retrieved by POLDER-3 the best, both in clear and cloudy atmospheres.

## 3.4 Aerosol optical properties in the WRF-Chem optimized configuration (S2)

Figure 14 illustrates the spectral dependencies of AOD (left) and SSA (right) simulated with the WRF-Chem optimized configuration (BCx2, OC/2.5 with 2.5 % of BrOC, red curves) and retrieved by POLDER-3/GRASP in clear-sky (black curves, top) and by POLDER-3/AERO-AC above clouds (black curves, bottom). Aerosol optical properties are averaged geographically on their respective studied areas (black frames in Figure 2) and temporally over the first half of July 2008.



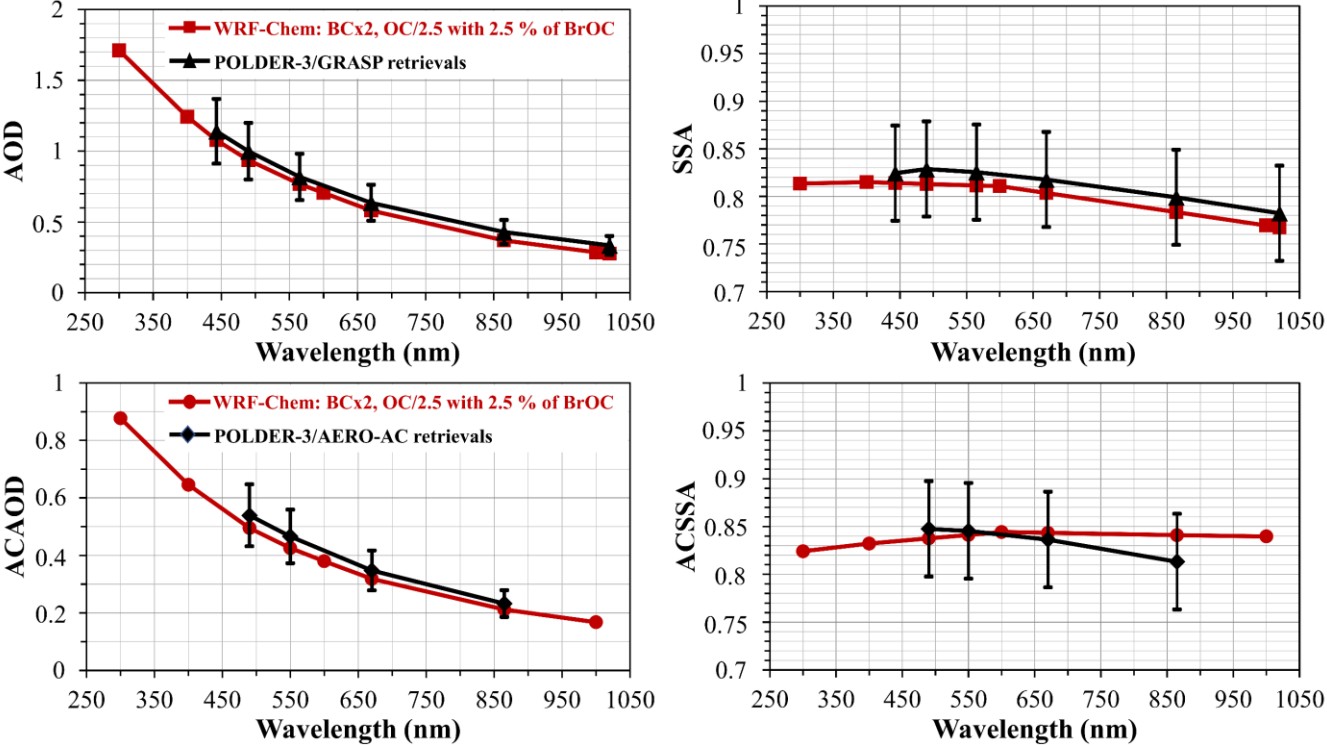

**Figure 14:** Spectral dependencies of AOD (left) and SSA (right) retrieved by POLDER-3/GRASP in clear-sky (top, black curves) and by POLDER-3/AERO-AC above clouds (bottom, black curves) and simulated with the WRF-Chem optimized configuration (BCx2, OC/2.5 with 2.5 % of BrOC, red curves). Black vertical error bars correspond to the uncertainties associated with POLDER-3 data with an accuracy of ±20 % on aerosol optical depth and ±0.05 on the single scattering albedo over the entire spectral domain. Values are averaged geographically on their respective study areas – (12°E-30°E, 10°S-0°S) over the Southern African continent in clear-sky and (0°E-15°E, 15°S-5°S) above clouds over the Southeastern Atlantic Ocean – and temporally over the first half of July 2008.

On Figure 14, we can see an increase of the AOD and the ACAOD spectral values with decreasing wavelength (left) and a decrease of the SSA and the ACSSA spectral values with increasing wavelength (right) retrieved by POLDER-3 (black curves in Figure 14). These trends are characteristic of fine mode aerosols containing BC, here BB (Dubovik et al. 2002; Bergstrom et al. 2007). Figure 14 shows a general good agreement between the WRF-Chem optimized configuration (red curves) and the POLDER-3 retrievals (black curves) on average for the first half of July 2008 with simulated spectral values of (AC)AOD and (AC)SSA included within the ranges of uncertainties associated with the POLDER-3 retrievals both in clear-sky and above clouds.

Over the Southern African continent in clear-sky (12°E-30°E, 10°S-0°S), the spectral values of the AOD are slightly underestimated in the WRF-Chem optimized configuration (red curve, top left) compared to those retrieved by POLDER-3/GRASP with an average bias of about -0.06 from visible to near-infrared. The spectral values of the SSA are also slightly underestimated in the WRF-Chem optimized configuration (red curve, top right) with an average bias of about -0.02 over the entire spectrum, reflecting a slightly higher aerosol absorption than that retrieved by POLDER-3/GRASP. This difference is





reduced at shorter wavelengths (average bias of -0.01 at 443 nm) due to the consideration of BrOC absorption (highly absorbing

especially in ultraviolet-blue, Kirchstetter et al. 2004; Hoffer et al. 2006) in the WRF-Chem optimized configuration (2.5 % of BrOC).

Above clouds over the SEAO (0°E-15°E, 15°S-5°S), the spectral values of the ACAOD are slightly underestimated in the WRF-Chem optimized configuration (red curve, bottom left) with an average bias ranging from -0.04 to -0.02 from 490 to 865 nm compared to those retrieved by POLDER-3/AERO-AC. Regarding the spectral behaviour of the ACSSA, the WRF-Chem

optimized configuration simulates an upward trend from 300 to 600 nm due to the presence of BrOC in the BBA plumes, then slightly decreasing in the near-infrared with increasing wavelength. However, the lack of data above clouds in the ultraviolet with POLDER-3/AERO-AC does not allow to assess the reliability of the increased in absorption below 490 nm due to the addition of BrOC to the WRF-Chem optimized configuration. In the near-infrared, the decrease in ACSSA values is more pronounced in the POLDER-3/AERO-AC retrieval with a positive average bias of 0.03 at 865 nm between the WRF-Chem

optimized configuration and the POLDER-3/AERO-AC retrieval. This difference could be due to the larger particles size simulated by WRF-Chem than those retrieved by POLDER-3 (see section 3.2.3).

Figure 15 illustrates the averaged $PM_{2.5}$ chemical composition simulated with the WRF-Chem optimized configuration, including organic aerosols (OA, in green), secondary inorganic aerosols (SIA, in grey), organic carbon (OC, in brown), black carbon (BC, in black), brown carbon (BROC, in purple), desert dust (DUST, in yellow) and sea salts (SEAS, in blue), and their

relative proportions over the Southern African continent in clear-sky (spatial average over 12°E-30°E, 10°S-0°S, left) and above clouds over the SEAO (spatial average over 0°E-15°E, 15°S-5°S, right) for the first half of July 2008.

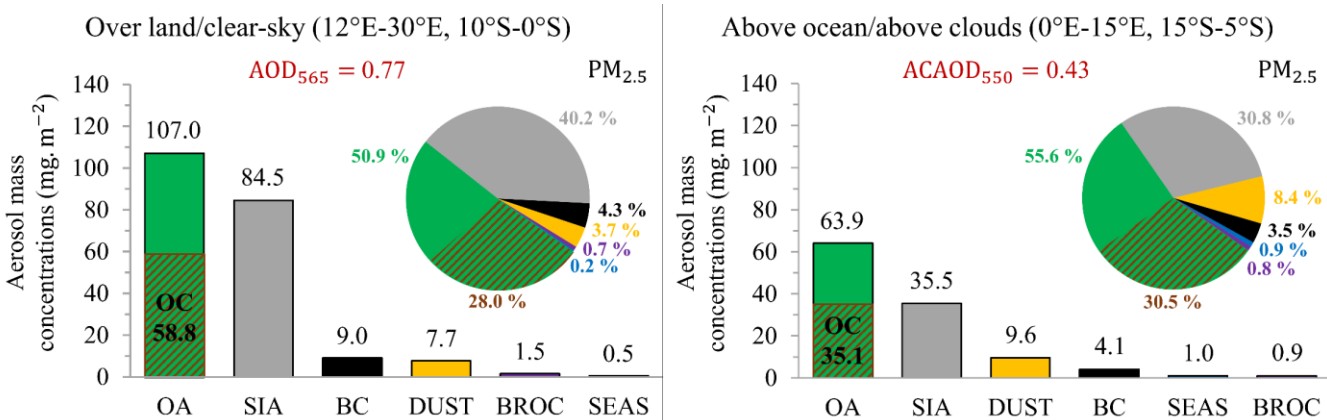

**Figure 15: Evolution of the $PM_{2.5}$ chemical composition simulated with the WRF-Chem optimized configuration between the Southern African continent in clear-sky (12°E-30°E, 10°S-0°S, left) and the Southeastern Atlantic Ocean above clouds (0°E-15°E,**
**15°S-5°S, right). Mass concentrations (in $mg.m^{-2}$) of the main aerosol compounds of biomass burning, i.e., black carbon (BC, in black), organic carbon (OC, in brown hatching), brown carbon (BrOC, in purple), organic aerosols (OA, in green) and secondary inorganic aerosols (SIA, in grey), as well as desert dust (DUST, in yellow) and sea salts (SEAS, in blue), are vertically integrated over the atmospheric column on average for the first half of July 2008.**

Figure 15 shows that the chemical composition of $PM_{2.5}$ simulated with the WRF-Chem optimized configuration still remains

realistic in comparison with the observations from previous field campaigns (Haslett et al. 2019; Denjean et al. 2020): organic



aerosols (OA, in green) still dominates, both at the BB sources (50.9 % of $PM_{2.5}$, left) and over the transport region (55.6 % of $PM_{2.5}$, right). Compared to the chemical composition of $PM_{2.5}$ simulated with the WRF-Chem reference configuration (see Figure 12), the contribution of black carbon (BC, in black) in the BBA plumes is greater in our studied area ($\geq 3.5$ %). The contribution of organic carbon (OC, brown hatching) is slightly lower: from 58.8 mg. $m^{-2}$ (28.0 % of $PM_{2.5}$) to 35.1 mg. $m^{-2}$

(30.5 % of $PM_{2.5}$) for WRF-Chem optimized configuration and from 67.0 mg. $m^{-2}$ (31.7 % of $PM_{2.5}$) to 45.4 mg. $m^{-2}$ (32.0 % of $PM_{2.5}$) for the WRF-Chem reference configuration, between sources and transport areas. As with the WRF-Chem reference configuration, the influence of desert dusts (DUST, in yellow) increases during the progression of the BBA plumes off the SEAO with a vertically integrated mass increasing on average from 7.7 mg. $m^{-2}$ (3.7 % of $PM_{2.5}$) to 9.6 mg. $m^{-2}$ (8.4 % of $PM_{2.5}$). The contribution of sea salts (SEAS, in blue) is negligible both over the Southern African continent (0.2 % of

$PM_{2.5}$) and above clouds over the SEAO (0.9 % of $PM_{2.5}$). Finally, the contribution of brown carbon (BrOC, in purple) is less than 1 % in our study area.

Figure 16 illustrates the spatial distributions of the monthly averaged AOD (top) and SSA (bottom) at 565 nm in clear-sky retrieved by POLDER-3/GRASP (left) and simulated with the WRF-Chem optimized configuration (right) for July 2008.

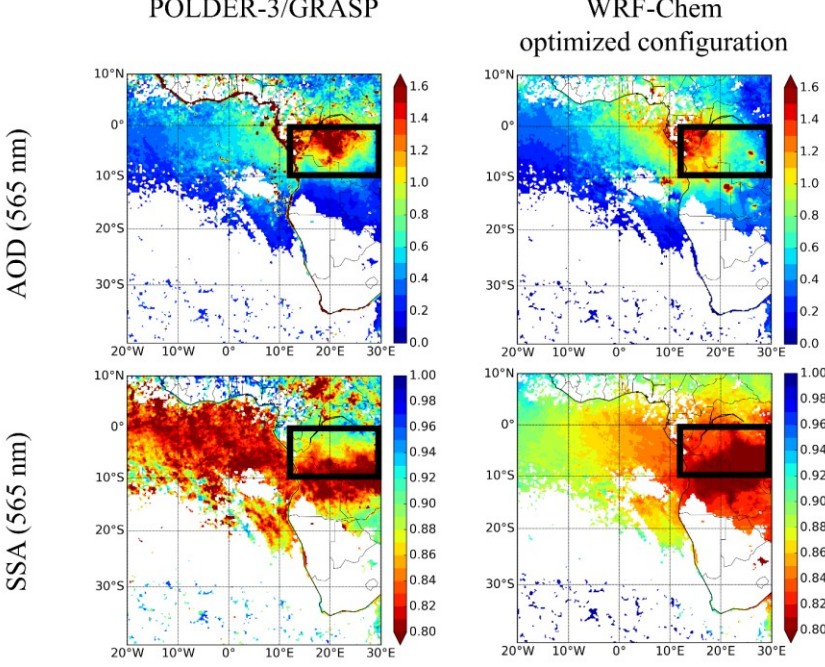

**Figure 16: Comparison of monthly averaged AOD (top) and SSA (bottom) at 565 nm in clear-sky retrieved by POLDER-3/GRASP (left) and simulated with the WRF-Chem optimized configuration (right) over the Southeast Atlantic region for July 2008. The black frames of geographic coordinates (12°E-30°E, 10°S-0°S) represent the area of study.**

Figure 16 shows that the results obtained with the WRF-Chem optimized configuration in clear-sky are overall satisfying on average for July 2008. Over the Southern African continent (12°E-30°E, 10°S-0°S), simulated mean AOD is 1.02 compared

to 1.10±0.22 at 565 nm for POLDER-3/GRASP, indicating a slight underestimation of the intensity of BBA emission sources





in the WRF-Chem optimized configuration. Their location is also somewhat different to the simulated emission sources located further west than those observed by POLDER-3/GRASP, resulting in a spatial correlation coefficient of 0.50 and an average bias of -0.08 at 565 nm. The simulated mean SSA is 0.81 compared to 0.85±0.05 at 565 nm for POLDER-3/GRASP showing a good agreement although the aerosol absorption is slightly overestimated in the WRF-Chem optimized configuration. Figure

16 also shows that the aerosol absorption is more uniform in the WRF-Chem optimized configuration in comparison to POLDER-3/GRASP that retrieves aerosols that are more scattering located north of the geographic coordinates area (12°E-30°E, 5°S-0°S). In this area, a contribution of other aerosol species to the BB plume is not excluded (mixture with desert dust aerosols) and could explain why POLDER-3 retrieves higher SSA than the model. This difference translates into a spatial correlation coefficient of 0.59 and an average bias of -0.04 at 565 nm. We can note that the mean SSA simulated with the

WRF-Chem optimized configuration at the Mongu, Zambia site (about 0.83 at 565 nm) in clear-sky in July 2008 is consistent with the one retrieved by AERONET (about 0.81 at 550 nm) at the same location during July 1997 to 2005 (Eck et al. 2013). Over the SEAO far from the continent, in-situ measurements from the ORACLES (Zuidema et al. 2018) and the CLARIFY-2017 (Taylor et al. 2020) airborne campaigns indicated a strong BBA absorption, consistently with the observations from POLDER-3/GRASP (mean SSA less than 0.80 at 565 nm). This may be due to coating and lens effects of BC, which appear

to be important for very old BBA plumes transported off the coast of Southern Africa (Zuidema et al. 2018; Denjean et al. 2020; Taylor et al. 2020; Wu et al. 2020). Remind that we do not consider coating in our approach and that, over this area, large discrepancies remain between POLDER-3 and the WRF-Chem optimized configuration.

Figure 17 illustrates the spatial distributions of monthly averaged ACAOD (top) and ACSSA (bottom) at 550 nm retrieved by POLDER-3/AERO-AC (left) and simulated with the WRF-Chem optimized configuration (right) for July 2008.

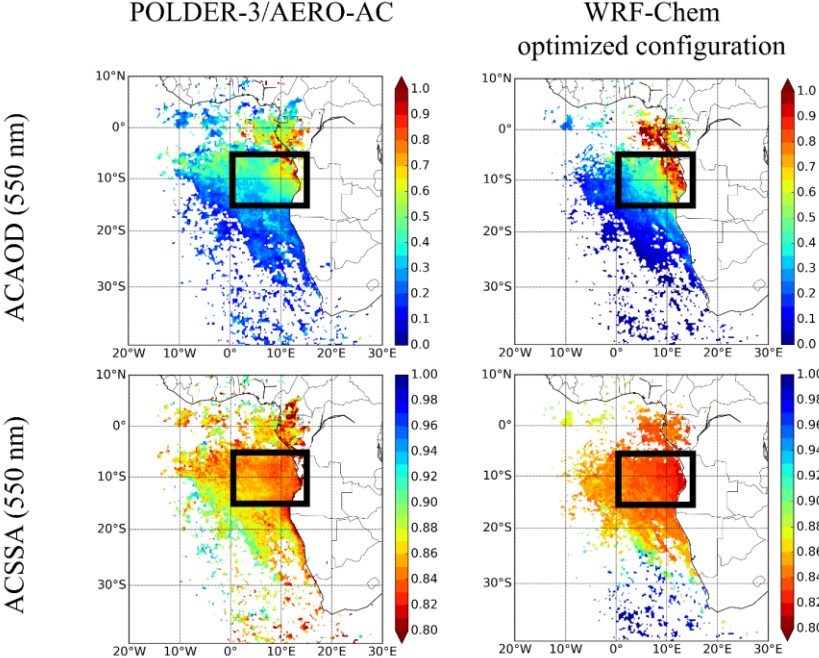


**Figure 17: Comparison of monthly averaged ACAOD (top) and ACSSA (bottom) at 550 nm above clouds retrieved by POLDER-3/AERO-AC (left) and simulated with the WRF-Chem optimized configuration (right) over the Southeast Atlantic region for July 2008. The black frames of geographic coordinates (0°E-15°E, 15°S-5°S) represent the area of study.**

Figure 17 shows a very good agreement between the WRF-Chem optimized configuration and the POLDER-3/AERO-AC

retrievals on the spatial distributions of both the ACAOD and the ACSSA at 550 nm averaged over July 2008. The simulated

mean ACAOD is 0.48 compared to 0.46±0.09 at 550 nm for POLDER-3/AERO-AC, indicating a good estimate of the amounts

of aerosols transported above clouds over the SEAO. Few minor differences can be observed within the studied area of

geographic coordinates (0°E-15°E, 15°S-5°S) resulting in a spatial correlation coefficient of 0.64 and an average bias of 0.02

at 550 nm. The simulated mean ACSSA is 0.84 compared to 0.85±0.05 at 550 nm for POLDER-3/AERO-AC, indicating a

good estimate of the aerosol absorption in the WRF-Chem optimized configuration. Some small differences can be observed

within the studied area of geographic coordinates (0°E-15°E, 15°S-5°S) leading to a spatial correlation coefficient of 0.66, an

average absolute error of only 0.01 and an average quadratic error of only 0.02.

In summary, the BBA absorption properties are significantly improved with the WRF-Chem optimized configuration while

maintaining a realistic content of aerosols transported above marine stratocumulus. Thus, the WRF-Chem optimized

configuration correctly simulates the content of BBA and their absorption properties over the SEA region on average for July

2008. In particular, the simulated mean SSA is consistent with the observed mean value of 0.85±0.02 over Southern Africa

(Leahy et al. 2007).



## 4 Conclusions and perspectives

This study shows that the WRF-Chem regional meteorological model coupled with chemistry combined with innovative
POLDER-3 satellite aerosol inversion algorithms provided a satisfying estimate of the load and absorption properties of BBA
over the SEA region, both in clear-sky and above clouds on average for July 2008.

The adjustment of the chemistry at the source level in the WRF-Chem optimized configuration (BCx2, OC/2.5 with 2.5%
BrOC) is consistent with the uncertainties present in APIFLAMEv1 BB emission inventory (Turquety et al. 2014) used in our
study. This aerosol chemical composition is obtained with the homogeneous internal mixing state. The mean BC/OC mass
mixing ratio simulated with this optimal scenario of WRF-Chem is estimated at 0.15 over the Southern African continent and
0.11 above clouds over the SEAO. These values are also in good agreement with previous in situ measurements over this
region, such as the SAFARI-2000 airborne campaign (Formenti et al. 2003). In addition, the absorbing fraction of OA in the
BBA plumes, i.e., BrOC, is estimated at 2-3 % for the studied period. This value could vary geographically and temporally.
Importantly, this BrOC contribution allows the increase in absorption retrieved by POLDER-3/GRASP at the shortest
wavelengths of the visible, characteristic of the presence of BrOC in the BBA plumes, to be well simulated with WRF-Chem
over the BB source areas. However, the lack of available POLDER-3 data for the SSA in the blue and in the ultraviolet above
clouds did not enable to constrain with certainty the BrOC content in cases of BBA plumes transported over stratocumulus off
the coasts of Namibia and Angola. This limitation is mainly related to the used inversion algorithm and may be exceeded in
future work. The simulated mean SSA are 0.81 (565 nm) and 0.84 (550 nm) in clear and above cloudy scenes respectively, in
good agreement with those retrieved by POLDER-3 (0.85±0.05 at 565 nm in clear-sky and at 550 nm above clouds) for the
studied period. Significant differences between modelled and measured SSA are only observed in remote ocean areas, 2000
km at least from the sources. These differences could be explained by the hypothesis of the homogeneous internal mixture that
we used in our study.

Our results highlight the fact that BC refractive index of Williams et al. (2007), which is more absorbing than that more
commonly used of Bond and Bergstrom (2006), provides the best estimates of BBA properties, allowing to reproduce the
retrievals from POLDER-3 satellite both in clear and cloudy skies. Other studies have shown that the Williams et al. (2007)
BC refractive index was more suitable for the assessment of the mass absorption cross section ($\text{MAC}_{\text{BC,fresh}} = 7.5 \pm$
$1.2\ \text{m}^2.\text{g}^{-1}$) associated with freshly emitted BC. In this sense, our work is consistent with the conclusions of the study on the
light absorption properties of BC from Liu et al. (2020).

The methodology developed during this study can be summarized as follows: using optimized regional simulations to constrain
the emission factors of big families of chemical species present in BB inventories through the adjustment of the chemistry.
This adjustment of the chemistry is made possible by the use of new satellite observations providing the SSA of aerosols and
its spectral dependence from space both in clear and cloudy skies. These recently available new observations used in synergy
with a regional model of meteorology coupled with chemistry, such as WRF-Chem, enable to obtain new constraints on the
aerosol chemistry in climate modelling exercises.





Our results can be considered as the first step of a larger study, which aims at providing more robust estimates of the climate impacts of BBA over the SEA region, including their interactions with clouds, using the optimized configuration of the WRF-Chem model. In particular, the influence of BrOC absorption on direct and semi-direct radiative effects of aerosols will be investigated. In addition, clouds response to contact with BBA plumes will be assessed at the indirect effects level.

In parallel, the methodology developed in this study could be applied throughout the summer 2008 dry season to evaluate the temporal variability and absorption cycle of BBA present in this region. Thus, an interesting development would be the elaboration of a parameterization of the BrOC content in WRF-Chem according to the evolution of fire activity during the dry season (Zuidema et al. 2016b; Zuidema et al. 2018; Pan et al. 2019; Pistone et al. 2019). To progress on this aspect, a synergy between satellite observations provided by the POLDER-3 and the OMI space sensors would allow to expand the retrieval of

the SSA of aerosols to the ultraviolet range, a spectral domain particularly sensitive to the presence of BrOC.

Another outlook will consist in applying our methodology to the more recent French AEROCLO-sA airborne campaign that took place in the SEA region in August-September 2017, using other satellite datasets than POLDER-3, for which observations ended in 2013. Other international measurement campaigns have also taken place in this region (ORACLES and CLARIFY airborne campaigns) in recent years (Zuidema et al. 2016b). They will provide detailed reference observations on the BBA

properties bringing additional constraints on their radiative and climate effects.

Finally, the methodology combining regional simulations and satellite observations of aerosols could be applied to other regions under the influence of BB, such as those located in the Northern Hemisphere or South America, particularly in the Amazon. Boreal forest fires are of interest because they are characterized by a very low carbonaceous mass mixing ratio and a slow combustion without flames. This type of fire is thus potentially richer in organic aerosols making it a potentially

important contributor of BrOC. For this type of plume, BrOC concentrations could therefore be much higher than for the BBA plumes observed over the SEA region, implying a potentially more pronounced BrOC effect on climate for these regions.

**Data availability**

The AERO-AC product is developed at the ICARE data and services centre (https://www.icare.univ-lille.fr) in Lille (France) in the frame of the POLDER/PARASOL mission and supported by CNES and PNTS (Waquet F., Peers F., Ducos F., Thieuleux

F., Deaconu L., A. Chauvigné and Riedi, J.: Aerosols above clouds products from POLDER/PARASOL satellite observations (AERO-AC products), doi:10.25326/82, 2020.).

The authors would like to acknowledge the use of POLDER data "POLDER/PARASOL Level-1 data originally provided by CNES (http://www.icare.univ-lille1.fr/) processed at Laboratoire d'Optique Atmosphérique with GRASP software (https://www.grasp-open.com) developed by Dubovik et al. (2011, 2014).



**Author contributions**

AS, FW, JCP and IC developed the concept of this paper. AS performed the WRF-Chem simulations and carried out the analyses of the POLDER and CALIOP data. ST developed the APIFLAME biomass burning emission inventory. FW and FP developed the POLDER-3/AERO-AC products and the data were processed by FT and FD. AS wrote the manuscript with contributions from all co-authors.

**Competing interests**

The authors declare that they have no conflict of interest.

**Acknowledgments**

This work has been supported by the University of Lille (France), the Programme National de Télédétection Spatiale (PNTS, http://www.insu.cnrs.fr/pnts, grant N° PNTS-2013-10), the French National Agency for Space Studies (CNES) and by the
CaPPA project (Chemical and Physical Properties of the Atmosphere), which is funded by the French National Research Agency (ANR) through the PIA (Programme d'Investissement d'Avenir) under contract « ANR-11-LABX-0005-01 » and by the Regional Council « Hauts-de-France » and the « European Funds for Regional Economic Development » (FEDER). The authors also thank the Région Hauts-de-France, and the Ministère de l'Enseignement Supérieur et de la Recherche (CPER Climibio), and the European Fund for Regional Economic Development for their financial support. The authors acknowledge
the AEROCLIM project of comparison of models and satellite data for climate studies, which is funded by CNES. The authors are also grateful to Oleg Dubovik and Pavel Litvinov for providing the GRASP data and their validation against AERONET data used in this study. The authors would like to thank the AEROCLO-sA team and project. The AEROCLO-sA project was supported by the ANR under grant agreement N° ANR-15-CE01-0014-01, the French national program LEFE/INSU, the Programme National de Télédétection Spatiale (PNTS, http://www.insu.cnrs.fr/pnts, grant N° PNTS-2016-14), and the South
African National Research Foundation (NRF) under grant UID 105958.

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
