# Peer review of "Combining POLDER-3 satellite observations and WRF-Chem numerical simulations to derive biomass burning aerosol properties over the Southeast Atlantic region"

_Atmospheric Chemistry and Physics, 2021_

## Referee Comment (RC2)

Review of: Combining POLDER-3 satellite observations and WRF-Chem numerical simulations to derive biomass-burins aerosol properties over the Southeast Atlantic region. Authors: A. Simeon, F. Waquet, J-C Pere, F. Docos, F. Thieuleux, F. Peers, S. Turquety, and I. Chiapello.

Manuscript number:

**Recommendation: Accept with Minor Revisions**

This is an interesting study, in which the authors delve into the representation of the SSA and AE in the WRF-Chem model, and use POLDER/GRASP retrievals to improve the model representations, and then examine the impact of changing OC/BC ratios, refractive indices, etc. in the model. It's a good idea and brings to the forefront the model representations, which can otherwise seem like a black box in publications. My comments are relatively small.

A main one is that I do not see any support for the idea that dust is present in the marine atmosphere in July. It's included in WRF-Chem, and the satellite retrievals produce a coarse-mode aerosol that could be dust or sea salt, but the authors do not authenticate its presence. Does CALIOP identify dust over the ocean? How about the Haslett or DenJean papers examining the southerly July flow? If the evidence for dust is slim, why not experiment with a model dust representation that excludes dust? I would suggest the authors do so if they cannot better support that the presence of dust is real.

An additional comment is that the authors underemphasize the present of BBA in the boundary layer. In July, a large percentage of the total BBA is in the boundary layer, as opposed to above the clouds. The Haslett and DenJean papers document this (as likely others coming out of DACCIWA although I am not as familiar with their literature) as do publications focused on Ascension Island (see, e.g., Zhang and Zuidema, 2021, ACPD and Zuidema et al 2018). ORACLES observations from the month of August also suggest this is likely the case, e.g., Kacarab 2020 ACP and Redemann ACP 2021.

Another comment is that the literature references did not make full use of the new results that have come out as a result of ORACLES/CLARIFY/LASIC/DACCIWA, and to the extent that they do, the references tend to be present later in the paper, as opposed to helping to establish the context within the introduction. I have listed some at the end of this, and either mention in the specific comments or as part of the references how I think they contribute.

A small comment is that the English in places sounds labored, using extra words that a native English speaker would leave out. I make a note of a few such spots below.

More minor/specific comments:

Title: the title is not entirely accurate I feel, as the study is more about using satellite retrievals to constrain the WRF-Chem aerosol representation. Perhaps the authors want to reconsider.

Introduction, lines33-35: worth noting is that southern Africa, which is the focus of this study, by itself produces one-third of the global annual carbon emissions from BB, according to the cited Werf paper.

Intro, line 38: Costantino and Breon 2013 is not really the right reference for documenting the aerosol transport. The African Easterly Jet-South is not yet active in July. Knippertz et al. 2017 might be a better fit for this, or references within.

Intro line 38-40: I am not sure the anticyclonic circulation responsible for long-range transport off of the continent is well established by July. Adebiyi and Zuidema 2016 suggests it isn't. And neither does Fig. 4. Fig. 4 does show an anticyclonic circulation in a couple of places but I don't see those affecting the regions selected for this study, shown in Fig. 2. Given that the authors have the model winds at their fingertips, perhaps they could say more about the circulations affecting their selected regions? It seems like the land domain might also be influenced by the west African monsoon? Do any of the cited papers discuss July? I am not sure they do.

introduction, 2nd paragraph, p.2: portions of this discussion feel dated, through the focus on the aerosol above clouds and neglect of the boundary layer BBA. Try to update.

p.3: This is a nice literature overview. I'm confused why the Denjean, Taylor, Pistone papers aren't included here. It's also a very long paragraph, could it be split into two?

p. 3 line 93: CLARIFY and ORACLES references should be included here.

p. 3 line 95: what does ANR stand for?

P.4 line 106: remove 'to perform'

P. 4 lines 115-118: would be nice to see more documentation of this, is this based on the authors' own analysis?

p. 7 fig 1: why include August and September? They are not used. I would suggest removing these panels.

p. 11 lines 291-292: on what basis do the authors believe that desert dust emissions from north Africa may significantly contribute to the total aerosol load?

p. 13, figure 4: is the ECMWF reanalysis the Interim analysis or ERA5?, also, the wind vectors are very difficult to read. Please replot with fewer and thicker vectors. It also seems to me, based on Fig. 9, that the fields at 500 hPa could be removed from Figures 4-6 without loss.

p. 14-15, lines 347-349: I cannot see winds capable of a westward aerosol transport in Fig. 4. I think the authors may be confusing the meteorology of September, which I suspect is what the cited papers focus on, with that of July.

p. 16 lines 384-386: given the finding that GOCART seems to raise too much dust, why not include an experiment in which its presence is reduced? (also 'dusts' -> 'dust')

p. 17 line 404: did the Koffi papers evaluate July explicitly? There is a strong seasonal cycle to the aerosol vertical structure, see, e.g. Redemann 2021, ACP

p. 18 line 435: "on the opposite" -> "In contrast"

p. 18 line 439: are the authors suggesting an aerosol invigoration effect on shallow clouds? This seems very unlikely to me and I see no reference. I would suggest just attributing the cloud paramterization, although it confuses me the parameterization cited (Lin) is a microphysical parameterization according to Table 1. Wouldn't the boundary layer scheme be the more likely cause?

p. 18 line 464: sea salt mixed with smoke I would think. What does the model say?

P. 18 lines 467-468: see also Shinozuka 2020 ACP, which shows many models share these aerosol layer altitude biases.

p. 19, fig. 10 left panel+discussion: is this for the free-tropospheric aerosol layer? An average over the full column? I'm confused by this, and how the coarse mode is increasing with distance. Incidentally the orange star is placed near Ascension Island, and some assessment could be done using the LASIC datasets if interested.

p. 22 lines 532-534: no mention of dust in the DACCIWA data description, further leading me to think an experiment should be done where it is removed from WRF-Chem and a further assessment done to see what additional changes have to be incorporated, for WRF-Chem to match the satellite retrievals.

p. 25 line 568: "an increased of the aerosol absorption' => 'increased aerosol absorption'

p. 25 line 573: "consistently' ->'consistent' here and elsewhere.

p. 28, line 643: is there any evidence for more desert dust becoming incorporated into the aerosol layer as it moves westward other than from the WRF-Chem model? Any observations of this?

p. 29 line 661: remove "that", 'Located" "the geographic coordinates area"

p. 30 line 692: is the Leahy 2007 representative of July? Eck 2013, Zuidema 2018 both show a strong seasonal evolution to the SSA so good to make sure it's about the same time frame.

p. 33 references: the formatting of the references is not consistent, check to make sure they fit the ACP format. It also seems like there are more references than are used?

**Additional reference suggestions:**

Adebiyi, A. and P. Zuidema, 2016: The role of the southern African easterly jet in modifying the southeast Atlantic aerosol and cloud environments. *Q. J. R. Meteorol. Soc.*, **142**, p. 1574-1589 doi: [10.1002/qj.2765]

Haywood, J. M., S. Abel, P. Barrett, et al, 2020: Overview: The CLoud-Aerosol-Radiation Interaction and Forcing: Year-2017 (CLARIFY-2017) measurement campaign, *Atmos. Chem. Phys.*, **21**, p. 1049-1084, doi:10.5194/ acp-21-1049-2021

Kacarab, M., et al, 2020: Biomass Burning Aerosol as a Modulator of Droplet Number in the Southeast Atlantic Region. *Atmos. Chem. Phys.*, **20**, p. 3029-3040, doi:10.5194/acp-20-3029-2020 - this focuses on the month of August, so slightly later than July, nevertheless supports the idea that in the early part of the BB season, much aerosol is also present in the boundary layer, and, indicates the microphysical implication.

Knippertz, P., Fink, A. H., Deroubaix, A., et al..: A meteorological and chemical overview of the DACCIWA field campaign in West Africa in June–July 2016, Atmos. Chem. Phys., 17, 10893–10918, https://doi.org/10.5194/acp-17-10893-2017, 2017

Redemann, J., R. Wood, P. Zuidema, et al, 2020: An overview of the ORACLES (ObseRvations of Aerosols above CLouds and their intEractionS) project: aerosol-cloud-radiation interactions in the Southeast Atlantic basin. *Atmos. Chem. Phys.*, **21**, p. 1507-1563, doi:10.5194/acp-21-1507-2021

Shinozuka, Y., P. E. Saide, G. A. Ferrada, et al. 2020: Modeling the smoky troposphere of the southeast Atlantic: a comparison to ORACLES airborne observations from September of 2016. *Atmos. Chem. Phys.*, **20**, p. 11,491-11,526, doi:10.5194/acp-20-11491-2020 - could be interesting (in a future study) to see how the new WRF-Chem representation does within this same comparison, the data are accessible in an easy meta format. Also useful for more context on where WRF-Chem falls in the pantheon of aerosol model representations. The downward 'slumping' for the aerosol layer shown in Fig. 9 is also shown to be common to many other aerosol modeling efforts.

J. Zhang and P. Zuidema, 2021: Sunlight-absorbing aerosol amplifies the seasonal cycle in low cloud fraction over the southeast Atlantic: *Atmos. Chem. Phys. Disc.*, doi:10.5194/acp-2021-275 - more information on July aerosol observations.

---

## Editor Decision (ED1)

This is a useful study, it's great to see the WRF-Chem interrogated to such an extent. The authors sufficiently addressed the reviewers' comments that I am not inclined to send the manuscript back to them, but I do have some comments I'd like to see addressed before this goes to publication. The following statements refer to the 'tracked changes' document:

Line 41: the statement that anticyclonic conditions occur 80% of the time in July needs more specificity. I see the authors mentioning the Swap and Garstang papers in the response to the reviewers as the source for this information, I'm not sure how those authors described the large-scale flow, but if you are going to go with this, at least include the citation and describe what they say with more detail. But keep in mind that July is the January of the southern hemisphere and such a land-based circulation is not likely to extend very far offshore. I don't see much of an anticyclonic circulation present in the figure 4. Zhang&Zuidema 2021 show above-cloud winds are weak at Ascension in July. Your nice fig. 7 also doesn't show evidence of a land-based anticyclonic circulation, the aerosol is just zonally diffused away from the continent. What would be more straightforward is to just say that there are prevailing if weak easterlies from 850-700hpa that advect aerosol westward. Your figures support that and then you'd be done.

Line 250 in tracked changes: 'against about'? Do you just mean 'about'?

Fig4: wind vectors difficult to see in the top 2 rows, can those be redone to be similar to the bottom two rows.

P.21 last 2 lines: this statement is simply incorrect. There is no anticyclonic circulation, at least not one over land, visible in fig. 4. This is the one I believe the cited papers would be referring to. The altitude of the smoke emission is low enough, especially in the model, that the anticyclonic circulation over the ocean, around the south Atlantic sea level pressure high, would have more of an influence in distributing the aerosol over the ocean than any land circulation. Again I think you can just say here that the winds blow westward off of land north of 20S.

P.25: an injection height of 6km seems high to me, your own caliop data shows the aerosol only extends up to 4km. But the aerosol advected over the ocean in the model is clearly too low. Can the authors comment? Does an injection height of 6km still mean that a lot of the aerosol is placed lower?

p. 28, fig. 9: this is a nice figure. We know that there is plenty of smoke in the boundary layer in July, from the Ascension Island measurements. CALIOP won't be able to discern this, but WRF-Chem might actually be doing better in the boundary layer than this figure would suggest. Would it be possible to show the WRF-Chem extinction from smoke or BC alone? Or alternatively could the authors mention the PC boundary layer smoke mass concentrations somewhere and place them in context with the Ascension Island rBC and, if desired, ACMS measurements of OA?

p. 31, line 715: presumably more of the long-range transport aerosol is in the boundary layer in the model, so the vertically-integrated volume size distribution will contain a larger contribution from the BL. This would suggest the coarse mode aerosol is mostly or entirely sea spray aerosol. Have the authors examined the AERONET data from Ascension? I also include micropulse lidar derived profiles of extinction and the depolarization ratio for July 2016 and 2017, these suggest some dust but it's not much. (extinction retrieved similarly to the AOD-constrained approach of MPLNET; https://agupubs.onlinelibrary.wiley.com/doi/10.1029/2018JD028867). It seems unlikely to me that there is more mass in dust than in sea spray, as stated by the authors on p. 34.

[Figure]

Fig. 12: doesn't seem consistent w ACSM values at Ascension. BC low compared to Z&Z2021.

P. 46 line 944: ORACLES->LASIC

P.50 line 1027: LASIC could also be mentioned here. Some of its data could have been used for this study, although what the authors have done is already interesting.

---

## Author Response (AR2)

RC1: 1. In this review, I'll reveal myself as a remote sensing specialist with less expertise in models such as WRF-Chem. With that in mind, I'd like to question the two step method of constraining WRF-Chem with POLDER and other data. Is this common practice? What is the benefit compared to constraining in one step, in other words adjusting WRF-Chem all at once based on POLDER and met data? I could understand differentiating between constraints that come from the POLDER retrievals and other data. But that is not the case – AOD from POLDER is used in step 1, microphysical properties in step 2. The problem is that the POLDER algorithms work by simultaneously retrieving all of those parameters, which I would think should be considered when using those data to adjust WRF-Chem. For example, if one were to impose an incorrect AOD in a POLDER retrieval, there would potentially be an incorrect retrieval of aerosol absorption (and vice versa). In a similar fashion, by initially modifying emissions to match AOD, and then later constraining aerosol optical properties might mean AOD no longer matches. I wonder if the POLDER/WRF-Chem match for AOD in Figure 16 and the ACAOD in Figure 17 could be further improved by constraining emissions a second time at this point. Which begs the question: why not just do it all at the same time. Am I missing something here?

AC: We agree with the reviewer that both AOD and SSA are linked and some changes on one parameter can simultaneously have an impact on the other. In our approach, we decided to constrain WRF-Chem in two steps to investigate separately the major factors impacting the AOD and then the SSA. The idea was to constrain first the AOD as this optical property is an extensive parameter, i.e., integrating all aerosols regardless of the origin of the source. Once this parameter well simulated, we could look further into the aerosol distribution by studying the sensitivity of the SSA to the chemical composition.

Thanks to this approach, we first evaluated the general performance of WRF-Chem in terms of aerosol emissions and transport modelling, i.e., the dynamics. We concluded that the WRF-Chem partially constrained configuration correctly simulated the general pattern of the transport of aerosols both in terms of location and intensity. Once the AOD constrained after adjusting the strength of biomass burning and desert dust aerosol emissions sources, we then studied the main microphysical properties influencing the SSA, i.e., the size, the chemical composition and the level of absorption of aerosols. We were thus able to determine an optimized configuration of WRF-Chem reproducing well the AOD and SSA both in clear and cloudy atmospheres for the studied period. Although it might be surprising, from a modelling point of view, checking and adjusting aerosol emissions is often an unavoidable preliminary step, because emissions inventories and schemes in models remain too uncertain, as highlighted in many publications (e.g., Turquety et al. 2014; Flaounas et al. 2016).

RC1: 1b. As an aside, I find the nomenclature for S1 "reference configuration" to be confusing. "Reference," to me, implies some external (to this effort) source of data, while instead S1 represents the first stage of constrained WRF-Chem results. Alternatives that may be better: "partially constrained configuration" or "emissions adjusted configuration" or something else more informative. Furthermore, Figure 3 could be made more useful by also listing what is constrained in WRF-Chem at each step.

AC: We appreciate your propositions. We replaced "reference configuration" by "partially constrained configuration" in the article. We modified Figure 3 by adding the investigated (in green color) and constrained (in red color) parameters in WRF-Chem (see below).

[Figure]

We modified its caption like "Figure 3: Schematic diagram of the general approach applied to constrain WRF-Chem simulations of aerosols using A-Train satellite coincident retrievals of the distributions of aerosols and clouds. AOD, ACAOD, SSA, ACSSA, $\sigma_{ext}$, AE, ACAE and $z_{CTH}$ stand for aerosol optical depth, above-cloud aerosol optical depth, single scattering albedo, above-cloud single scattering albedo, aerosol extinction coefficient, Angstrom exponent, above-cloud Angstrom exponent and cloud top height.".

RC1: 2. Continuing the discussion on constraint methodology: why is it so approximate? In section 3.2.1 you adjust the APIFLAMEv1 BB emissions inventory by a factor of 1.5, while the ratio of WRF-Chem to POLDER-3/GRASP AOD(565) is 0.49/1.10 = 2.24. I imagine one reason could be that AOD does not directly relate to emissions estimates, but a discussion of how you go from that ratio value to 1.5 is needed. The same could be said for the choice of the Williams et al. (2007) BC refractive index, BCx2, OC/2.5 and 2.5% BrOC: is this the best match you can make? Could you do some more fine tuning to get a better constraint, or are the computational needs too much?

AC: Indeed, the AOD does not only relate to emissions estimates, but also on all the physico-chemical processes influencing the aerosol population. We first applied a multiplying factor of 2 to the APIFLAMEv1 BB emissions inventory and the result was not concluding. Over land (12°E-30°E, 10°S-0°S), simulated AOD values ($AOD_{mod,565} = 0.98$) were in much better agreement with those retrieved by POLDER-3/GRASP ($AOD_{obs,565} = 1.10$), with a moderate underestimation of 0.12 on a geographical average for July 2008. Nevertheless, above the ocean, simulated AOD values ($AOD_{mod,565} = 0.86$) were overestimated compared to POLDER-3/GRASP retrievals ($AOD_{obs,565} = 0.61$), with a positive bias of 0.25 on a geographical average for July 2008. This could be explained by a higher emitted and then transported aerosol load. This could also be interpreted as a numerical diffusion problem of a dense biomass burning aerosol (BBA) plume, which causes a spread of the load between the continent and the ocean due to the width of the mesh of WRF-Chem (as a reminder, a grid of 30x30 km²). Finally, the adjustment factor of 1.5 was chosen as being a good compromise, even if small biases were still present. This point is now clarified in the revised manuscript.

To our knowledge, the mass absorption cross section (MAC) associated with the Williams et al. (2007) BC refractive index, $MAC_{BC,fresh} = 7.2 \, m^2.g^{-1}$, is the closest to the reference value of $MAC_{BC,fresh} = 7.5 \pm 1.2 \, m^2.g^{-1}$ (Bond and Bergstrom, 2006) from the literature. Therefore, it seemed interesting to include this refractive index in our sensitivity study.

Our sensitivity study, including 96 WRF-Chem experiments, took several months to run and to analyze results. The final result (BCx2, OC/2.5 and 2.5% BrOC) might not be the finest result but represents our best match.

As explained in the revised manuscript, a score is computed for each WRF-chem numerical simulation. The score is equal to the number of times the result of the simulation is included within the range of the retrieval uncertainties. This score is computed over all the pixels for the two selected areas (with and without clouds) and over all available spectral aerosol parameters (i.e., AOD, SSA, 4 spectral bands for AERO-AC and 6 for GRASP). As a result, with the BC refractive index of Bond and Bergstrom (2006), we were unable to find a realistic solution of WRF-Chem within the uncertainties of APIFLAMEv1 (Turquety et al. 2014). With the BC refractive index of Williams et al. (2007), this optimized configuration of WRF-Chem was obtained.

2b. By the way, I think table 4 would be more informative if the 'score' you used in section 3.3 were also included. And regarding how that score is calculated: why not use statistically appropriate parameters such as the Mean Absolute Error (MAE) or Root-Mean Squared Error (RMSE)?

AC: This is an interesting suggestion but we kept Table 4 as it for clarity and conciseness. Indeed, it would require to add three more lines per numerical experiment, the scores being different for each proportion of brown carbon. The scores that we calculated allowed us to establish a ranking of the numerical simulations in descending order, no numerical simulation having obtained exactly the same score. In addition, the higher the score, the more the bias is reduced between the WRF-Chem numerical simulation and the POLDER-3 retrievals. This method of calculation being sufficient, we were able to do the ranking without using these statistical parameters (MAE and RMSE).

RC1: 3. Another issue relates to how you visualize and assess the differences between POLDER and the various versions of WRF-Chem. I like the images in, for example, Fig. 7, 8, 11, etc. since they give a good intuitive understanding of the spatial context in comparing the two. However, it is qualitative. Since you've aggregated the POLDER data to the coarser WRF-Chem model grid, you have the ability to do a pixel by pixel comparison – in which case you can calculate and visualize more statistics. In most cases, you present a mean difference between POLDER and WRF-Chem, when instead you could show a histogram of differences, and make a mean-bias ("Bland-Altman") or some other sort of plot. Please just don't use the (unfortunately common) scatterplot and calculate correlation coefficients, as that would be statistically inappropriate (see for example, Altman and Bland, 1983, https://doi.org/10.2307/2987937). One of the reasons this matters is because AOD (and presumably ACAOD) is not normally distributed (Sayer and Knobelspiesse, 2019, https://doi.org/10.5194/acp-19-15023-2019).

AC: Thank you for this interesting suggestion. As suggested, we included a statistical part with histograms to Fig. 8, for example, in the revised manuscript.

[Figure]

**Figure 8: Monthly averaged cloud top heights retrieved by POLDER-3 (top, left) and simulated with the WRF-Chem partially constrained configuration with the threshold of 95 % (top, right) for July 2008, with an associated statistical analysis of the threshold values of the cloud top height tested in the model (bottom). The black frames (0°E-15°E, 15°S-5°S) represent the study area.**

This revised Fig. 8 now better helps to understand the choice of the threshold of 95 % in the WRF-Chem partially constrained configuration.

RC1: 4. Essentially, you're only assessing half of one month of one year. I realize that many of your choices are driven by computational resources, but how do you expect these results to fit within the annual cycle of BB in southern Africa, and how relevant is it considering inter-annual variability? I was expecting some discussion on this. These results need to be expressed in terms of their general usefulness. Perhaps a few words on computation resources needed for this analysis would be useful too, should somebody want to scale up to a full year or years.

AC: We extended the study period until September 30, 2008 with the WRF-Chem optimized configuration to test its ability to represent the absorption cycle observed during the fire season of 2008. Figure A shows the temporal evolutions of the ACAOD (left) and ACSSA (right) retrieved by POLDER-3/AERO-AC (solid line curves) and simulated with the WRF-Chem optimized configuration (dotted curves) above clouds at 490, 550 and 865 nm (blue, green and red curves, respectively), during the July-August-September 2008 quarter.

[Figure]

**Figure A: Temporal evolutions of the ACAOD (left) and the ACSSA (right) retrieved by POLDER-3/AERO-AC (solid line curves) and simulated with the WRF-Chem optimized configuration (BCx2, OC/2.5 with 2.5% of BrOC, dotted curves) above clouds over the Southeast Atlantic Ocean at 490, 550 and 865 nm (blue, green and red curves, respectively). The optical properties are averaged geographically within the coordinate study area of (0°E-15°E, 15°S-5°S) and monthly over the July-September 2008 period. The vertical error bars correspond to the uncertainties associated with POLDER-3 data with an accuracy of ±20 % (per pixel estimate) on the aerosol optical depth and ±0.05 (per pixel estimate) on the single scattering albedo over the entire spectral domain.**

On the one hand, Figure A shows that the WRF-Chem optimized configuration reproduces in part at least the increase in ACAOD retrieved by POLDER-3/AERO-AC between July and August, but does not reproduce the variability observed between August and September 2008. For September 2008, the WRF-Chem optimized configuration fails to adequately simulate the aerosol concentration peak which is typically observed at this time of year over the Southeast Atlantic region. Indeed, the WRF-Chem optimized configuration underestimates the amounts of aerosols transported over the marine stratocumulus clouds with an average bias of -0.24 for ACAOD at 550 nm compared to the POLDER-3/AERO-AC retrievals in September 2008. This underestimation could be related to uncertainties about the evolution of biomass burning emissions during the season in the inventory used (APIFLAMEv1, Turquety et al. 2014). Recently, a numerical sensitivity study was conducted by Pan et al. (2019) testing six global biomass burning emissions inventories (excluding APIFLAMEv1) in the NASA's GEOS-Chem chemistry-transport model. In particular, this modelling study focused on the South African continent in September 2008. The results of the study showed an underestimation of the simulated AODs over land for these six inventories, with a maximum bias of 0.23 compared to MODIS-Aqua data at 550 nm and at least 50% compared to data from the AERONET site in Mongu, Zambia. This discrepancy between modelling and observations could not be explained by different synoptic conditions because the meteorology and aerosol mechanisms (transport and removal) were identical for these six numerical experiments. The authors of this study therefore suggest that estimates of the amounts of aerosols emitted by biomass burning should be revised high, in particular by updating the emission factors according to the season and the conditions of activity of the fires.

On the other hand, Figure A shows that the WRF-Chem optimized configuration struggles to represent very satisfyingly the seasonal cycle of ACSSA retrieved by POLDER-3/AERO-AC, which is characterized by a decrease in aerosol absorption (increase in ACSSA values) during the advance of the fire season. This result suggests that the chemical composition of the BBA plumes, especially the carbonaceous aerosols, changes during the dry season, likely in relation to a change in the type of fuel burned and in combustion conditions (Zuidema et al. 2016b; Pan et al. 2019). It is therefore necessary to study the aerosols physico-chemical and optical properties

on a month-by-month basis in order to better understand and constrain the BBA absorption cycle over the South-East Atlantic region in WRF-Chem.

This point is now discussed in more details in the conclusion section of the revised manuscript. We added the following sentences at line 127, page 32: "First results show that the WRF-Chem optimized configuration struggles to represent very satisfyingly the seasonal cycle of ACSSA retrieved by POLDER-3/AERO-AC, which is characterized by a decreased aerosol absorption (increase in ACSSA values) during the advance of the fire season. This suggests that the chemical composition of the BBA plumes, especially the carbonaceous aerosols, changes during the dry season, likely in relation to a change in the type of fuel burned and in combustion conditions (Zuidema et al. 2016b; Zuidema et al. 2018; Pan et al. 2019; Pistone et al. 2019).".

RC1: 5. Is the RETRO anthropogenic emissions inventory, which doesn't cover the year of your analysis, the best source for that information?

AC: We chose the RETRO global anthropogenic emissions inventory for its higher temporal resolution (monthly) than other global anthropogenic emissions inventories (generally yearly) such as EDGARv4.2 (Janssens-Maenhout et al. 2010). We agree with the reviewer that RETRO does not cover the specific year of our analysis, but this long-term climatology still remains representative of the anthropogenic influence during the African dry season.

RC1: 6. When considering POLDER measurement uncertainty, please recall that this often refers to per pixel estimates. In that sense, are the error bars in Figure 14 appropriate, since they are applied to geographically averaged values?

AC: The reviewer is right that uncertainties are for each pixel. In our approach, only the data with the lowest uncertainties are kept and these uncertainties (±15% for AOD and ±0.05 for SSA) are the same for each pixel. Hence the averaging procedure does not change our results. This point is now clarified in the revised manuscript.

RC1: More minor points:

RC1: 1. Why are the reference wavelengths for SSA and AOD different for the POLDER/GRASP and above cloud products different? I agree that they're close enough to not matter, but it should be explained.

AC: The POLDER-3/AERO-AC product does not use the wavelength at 565 nm because it is not a polarized channel. For the POLDER-3/AERO-AC retrievals, the 550 nm channel was added as a request of modelers who typically use this wavelength in the visible range for the evaluation of simulated aerosol parameters. Considering that the absorption of aerosols does not depend on the wavelength between 490 and 865 nm in POLDER-3/AERO-AC, the aerosol optical properties can be calculated at 550 nm with the Mie's theory, knowing the size of the particles, their absorption and their optical thickness (at 865 nm).

RC1: 2. Page 2, line 45: Clarify if "BBA direct radiative forcing" refers or global or SEAO.

AC: "BBA direct radiative forcing" refers to the SEAO. We added "Over the SEAO" at the beginning of the sentence, line 45, page 2.

RC1: 3. Page 14, line 125: I think you mean "specialty" not "specificity".

AC: Indeed, we mean "specialty".

RC1: 4. Table 2 and discussion in text. I'm a little confused what you mean by "range of uncertainties" here. Does this really mean range of the values reported in the literature?

AC: You are right. This really means range of the values reported in the literature. We replaced "range of uncertainties" by "range of values" in the article.

RC1: 5. Page 21, lines 507-8: Is the really the primary source of the differences, or could it also be due to the nature of your WRF-Chem constraint as well?

AC: The SSA of aerosols depends on both their size and chemical composition. In section 3.2.3 entitled "Particle size", we checked that the size of aerosols simulated by WRF-Chem is realistic in comparison with the POLDER-3 retrievals, without any modification of the physico-chemical processes in the model. Furthermore, the relative humidity is another important factor that may impact the SSA. In section 3.1 entitled "Meteorology in the WRF-Chem initial configuration (S0)", we ensured that the humidity is correctly simulated by WRF-Chem in comparison with the ECMWF reanalysis, with spatial correlation coefficients higher than 0.93 and very low mean errors, mean absolute errors and root mean square errors values. Therefore, we made the assumption that the chemical composition of aerosols is the primary source of the differences on the SSA between WRF-Chem and the POLDER-3 retrievals. Finally, one important key parameter included in the aerosol chemical composition is the aerosol mixing state. In our study, we only considered the homogeneous internal mixing state of particles in our WRF-Chem numerical simulations as in the POLDER-3 satellite inversion algorithms. The core-shell configuration, which appears to be important for very old BBA plumes transported off the coast of Southern Africa (Zuidema et al. 2018; Denjean et al. 2020; Taylor et al. 2020; Wu et al. 2020), could also have an influence on BBA optical properties and it would deserve to be tested.

RC1: 6. Page 25, line 577: Is the score calculated for AOD, SSA, AND the top of atmosphere spectral observations? If so, the latter are used to derive the former, so the logic seems a bit circular.

AC: The score is calculated for the AOD($\lambda$) and the SSA($\lambda$) over the clear-sky area (see Fig. 2, left) based on the POLDER-3/GRASP retrievals, and for the ACAOD($\lambda$) and the ACSSA($\lambda$) over the cloudy area (see Fig. 2, right) based on the POLDER-3/AERO-AC retrievals.

RC1: 7. Page 29, line 657: You mention here and elsewhere a "spatial correlation coefficient" but give no description of what you mean by this or how it is calculated. Similarly on Page 30, line 687. Are those error metrics what I'm calling for in 3? Regardless, they need to be explained.

AC: Thank you for this suggestion. This point is now clarified in the revised manuscript. Those errors metrics are the statistical parameters you are calling for in question 3. We added an appendix A to the revised manuscript describing the statistical parameters that we used in our study. We also added the following sentence at line 331, page 12: "The performance of the WRF-Chem numerical simulations will be assessed using commonly used statistical parameters (see Appendix A) (Thunis et al. 2011; Lingard et al. 2013): the Pearson correlation coefficient (R), the mean bias (MB), the mean absolute error (MAE) and the root mean square error (RMSE)."

**Appendix A: Description of the statistical parameters**

The Pearson correlation coefficient (R) is calculated as follows:

$$R = \frac{\sum_{i=1}^{N}(M_i - \overline{M})\,(O_i - \overline{O})}{\sqrt{\sum_{i=1}^{N}(M_i - \overline{M})^2}\,\sqrt{\sum_{i=1}^{N}(O_i - \overline{O})^2}}$$

$M_i$, $O_i$, $\overline{M}$, $\overline{O}$ et $N$ are respectively the simulated value, the observed value, the average of the simulated and observed values, and the total number of pixels. R is between -1 and +1, with a value tending towards zero indicating no linear correlation.

The mean bias (MB) is the average of the difference between a simulated and an observed value over an area or over a specified period:

$$MB = \frac{\sum_{i=1}^{N}(M_i - O_i)}{N}$$

If the difference between the simulated value and the observed value is non-zero then the simulated value is said to be biased, positively in the case of overestimation or negatively otherwise.

The mean absolute error (MAE) is calculated from the absolute value of the difference between a simulated and an observed value:

$$MAE = \frac{\sum_{i=1}^{N}|M_i - O_i|}{N}$$

This quantity is therefore always positive and tends towards zero when the simulated values are close to those observed. This statistical parameter is more restrictive than the average bias because it avoids possible error compensations.

The root mean square error (RMSE) is calculated from the square root of the root mean square difference between the simulated and observed values:

$$RMSE = \sqrt{\frac{\sum_{i=1}^{N}(M_i - O_i)^2}{N}}$$

Point but large biases can produce a high RMSE value. This statistical parameter is therefore commonly used as a measure of the overall performance of a model.

AC: References

Bond, Tami C., et Robert W. Bergstrom. 2006. « Light Absorption by Carbonaceous Particles: An Investigative Review ». Aerosol Science and Technology 40 (1): 27-67. https://doi.org/10.1080/02786820500421521.

Denjean, Cyrielle, Thierry Bourrianne, Frederic Burnet, Marc Mallet, Nicolas Maury, Aurélie Colomb, Pamela Dominutti, et al. 2020. 'Overview of Aerosol Optical Properties over Southern West Africa from DACCIWA Aircraft Measurements'. Atmospheric Chemistry and Physics 20 (8): 4735–56. https://doi.org/10.5194/acp-20-4735-2020.

Flaounas, Emmanouil, Vassiliki Kotroni, Konstantinos Lagouvardos, Martina Klose, Cyrille Flamant, and Theodore M. Giannaros. 2016. 'Assessing Atmospheric Dust Modelling Performance of WRF-Chem over the Semi-Arid and Arid Regions around the Mediterranean'. Atmospheric Chemistry and Physics Discussions, May, 1–28. https://doi.org/10.5194/acp-2016-307.

G. Janssens-Maenhout, A. M.R. Petrescu, M. Muntean & V. Blujdea (2011) Verifying Greenhouse Gas Emissions: Methods to Support International Climate Agreements, Greenhouse Gas Measurement and Management, 1:2, 132-133, DOI: 10.1080/20430779.2011.579358

Lingard, Justin, Lorenzo Labrador, Daniel Brookes, et Andrea Fraser. 2013. « Statistical Evaluation of the Input Meteorological Data Used for the UK Air Quality Forecast (UK-AQF) », no 1: 34.

Pan, Xiaohua, Charles Ichoku, Mian Chin, Huisheng Bian, Anton Darmenov, Peter Colarco, Luke Ellison, et al. 2019. 'Six 1100 Global Biomass Burning Emission Datasets: Inter-Comparison and Application in One Global Aerosol Model'. Preprint. Aerosols/Atmospheric Modelling/Troposphere/Physics (physical properties and processes). https://doi.org/10.5194/acp-2019-475.

Taylor, Jonathan W., Huihui Wu, Kate Szpek, Keith Bower, Ian Crawford, Michael J. Flynn, Paul I. Williams, et al. 2020. 'Absorption Closure in Highly Aged Biomass Burning Smoke'. Atmospheric Chemistry and Physics 20 (19): 11201–21. https://doi.org/10.5194/acp-20-11201-2020.

Thunis, P., Emilia Georgieva, et Stefano Galmarini. 2011. A procedure for air quality models benchmarking, Version 2.

Turquety, S, Laurent Menut, B Bessagnet, Alessandro Anav, Nicolas Viovy, Fabienne Maignan, and M Wooster. 2014. APIFLAME v1.0: High-Resolution Fire Emission Model and Application to the Euro-Mediterranean Region. Vol. 7. https://doi.org/10.5194/gmd-7-587-2014.

Williams, T. C., C. R. Shaddix, K. A. Jensen, et J. M. Suo-Anttila. 2007. « Measurement of the dimensionless extinction coefficient of soot within laminar diffusion flames ». International Journal of Heat and Mass Transfer 50 (7): 1616-30. https://doi.org/10.1016/j.ijheatmasstransfer.2006.08.024.

Wu, Huihui, Jonathan W. Taylor, Kate Szpek, Justin M. Langridge, Paul I. Williams, Michael Flynn, James D. Allan, et al. 2020. 'Vertical Variability of the Properties of Highly Aged Biomass Burning Aerosol Transported over the Southeast Atlantic during CLARIFY-2017'. Atmospheric Chemistry and Physics 20 (21): 12697–719. https://doi.org/10.5194/acp-20-12697-2020.

Zuidema, Paquita, Jens Redemann, James Haywood, Robert Wood, Stuart Piketh, Martin Hipondoka, and Paola Formenti. 2016b. 'Smoke and Clouds above the Southeast Atlantic: Upcoming Field Campaigns Probe Absorbing Aerosol's Impact on Climate'. Bulletin of the American Meteorological Society 97 (7): 1131–35. https://doi.org/10.1175/BAMS-D-15-00082.1.

Zuidema, Paquita, Arthur J. Sedlacek, Connor Flynn, Stephen Springston, Rodrigo Delgadillo, Jianhao Zhang, Allison C. Aiken, Annette Koontz, and Paytsar Muradyan. 2018. 'The Ascension Island Boundary Layer in the Remote Southeast Atlantic Is Often Smoky'. Geophysical Research Letters 45 (9): 4456–65. https://doi.org/10.1002/2017GL076926.

RC2: Review of: Combining POLDER-3 satellite observations and WRF-Chem numerical simulations to derive biomass-burning aerosol properties over the Southeast Atlantic region.

RC2: Authors: A. Simeon, F. Waquet, J-C Pere, F. Docos, F. Thieuleux, F. Peers, S. Turquety, and I. Chiapello.

RC2: Manuscript number:

RC2: Recommendation: Accept with Minor Revisions

RC2: This is an interesting study, in which the authors delve into the representation of the SSA and AE in the WRF-Chem model, and use POLDER/GRASP retrievals to improve the model representations, and then examine the impact of changing OC/BC ratios, refractive indices, etc. in the model. It's a good idea and brings to the forefront the model representations, which can otherwise seem like a black box in publications. My comments are relatively small.

RC2: A main one is that I do not see any support for the idea that dust is present in the marine atmosphere in July. It's included in WRF-Chem, and the satellite retrievals produce a coarse mode aerosol that could be dust or sea salt, but the authors do not authenticate its presence. Does CALIOP identify dust over the ocean? How about the Haslett or Denjean papers examining the southerly July flow? If the evidence for dust is slim, why not experiment with a model dust representation that excludes dust? I would suggest the authors do so if they cannot better support that the presence of dust is real.

AC: We investigated the vertical profiles of the modeled main aerosol volume concentrations and the aerosol extinction coefficient at 550 nm along the transport of the biomass burning aerosol (BBA) plume over the Southeast Atlantic (SEA) region averaged over July 2008. We plotted the evolution of the chemical composition of carbonaceous aerosols (BC in black and OC in brown), DUST (in yellow) and SEAS (in blue) and the aerosol extinction at 550 nm (COEFF_EXT_550, dotted red curves) over the whole atmospheric column along the transport of the BBA plume over the SEA region, as shown in Fig. A (not shown in the article, see below).

[Figure]

**Figure A: Vertical profiles of the aerosol extinction coefficient at 550 nm ($\sigma_{ext}$ in $km^{-1}$, red dotted curves) and the volume concentrations (in $\mu g.\,m^{-3}$) of carbonaceous aerosols, i.e. black carbon (BC, black solid curves) and organic carbon (OC, brown solid curves), desert dust (DUST, yellow solid curves) and sea salt (SEAS, blue solid curves) simulated by the WRF-Chem reference configuration along the path of the biomass burning aerosol plume, from its emission to its long-distance transport over the Southeast Atlantic region, on average for July 2008. Vertical profiles are plotted for the pixels in Fig. 10 respectively for emission sources (left, blue star), halfway transport (middle, red star) and long-range transport (right, orange star).**

Figure A firstly shows that the altitude of the BBA plume gradually decreases with a maximum of the aerosol volume concentration located at around 1.5 km altitude (halfway transport) and then 500 m altitude (long-range transport) over the Southeastern Atlantic Ocean. Figure A secondly shows that volume concentration of sea salt aerosols (SEAS, blue solid curves) and desert dust aerosols (DUST, yellow solid curves) gradually increase between the emission sources and the transport of the BBA plume off the African southern coasts. Besides, variations in the aerosol extinction coefficient at 550 nm (COEFF_EXT_550, dotted red curves) appear to be little influenced by the presence of mineral dust from the emission to the removal of the BBA plume. For example, the peak concentration of desert dust aerosols located at about 2.5 km altitude (yellow curve) is not associated with a significant increase in the aerosol extinction coefficient at 550 nm (red dotted curve) at the level of the long-range transport of the BBA plume. This would suggest that the influence of desert dust aerosols on the aerosol optical properties appears weak at 550 nm at least unlike carbonaceous aerosols (black carbon and organic carbon) constituting the BBA plume. We concluded desert dust aerosols are little present within the BBA plume but they become predominant as the BBA plume moves westward and dissipates very far from the southern African coasts.

This characteristic is supported by the recent work of Deaconu et al. (2019) over the same region using a set of satellite observations from the A-Train constellation. They showed, for the 2006-2009 period (including July months), the presence of dust transported above clouds off the coasts of the Southeast Atlantic Ocean (SEAO). The work of Denjean et al. (2020) also confirmed the presence of mineral dust transported from Sahara and Sahel in the norther part of our domain (e.g., near the coasts of Ghana and Ivory coast).

In the paper, after the following sentence in page 11 at lines 291-292: "Desert dust emissions from North African sources are also considered as they may contribute to the total aerosol load over our studied area.", we added the

following sentence: "The transport of mineral dust off the coast of the SEAO (see figure 1) is notably supported by recent satellite and airborne observations performed in the northern part of our domain (Deaconu et al. 2019, Denjean et al. 2020)."

RC2: An additional comment is that the authors underemphasize the present of BBA in the boundary layer. In July, a large percentage of the total BBA is in the boundary layer, as opposed to above the clouds. The Haslett and Denjean papers document this (as likely others coming out of DACCIWA although I am not as familiar with their literature) as do publications focused on Ascension Island (see, e.g., Zhang and Zuidema, 2021, ACPD and Zuidema et al 2018). ORACLES observations from the month of August also suggest this is likely the case, e.g., Kacarab 2020 ACP and Redemann ACP 2021.

AC: The reviewer is right and this point is now mentioned in the revised manuscript by adding the following sentence at line 60, page 2: "It is worth noting that the transport of BBA in the boundary layer has also been reported by recent airborne observations (Zuidema et al. 2018, Haslett et al. 2019, Denjean et al. 2020, Kacarab et al. 2020, Redemann et al. 2021, Zhang and Zuidema, 2021).".

RC2: Another comment is that the literature references did not make full use of the new results that have come out as a result of ORACLES/CLARIFY/LASIC/DACCIWA, and to the extent that they do, the references tend to be present later in the paper, as opposed to helping to establish the context within the introduction. I have listed some at the end of this, and either mention in the specific comments or as part of the references how I think they contribute.

AC: This is an interesting remark. The bibliography has been now updated in the revised manuscript by adding most of the recent papers you mentioned. An outlook of the current study could consist in applying our methodology to the 2016-2018 period, which corresponds to the recent observation campaigns (AEROCLO-sA, ORACLES, CLARIFY, LASIC and DACCIWA), to take full advantage of the new results on the BBA properties and additional constraints on their radiative and climate effects.

RC2: A small comment is that the English in places sounds labored, using extra words that a native English speaker would leave out. I make a note of a few such spots below.

AC: We would like to thank you for your comments which are helpful to improve the English.

RC2: More minor/specific comments:

RC2: Title: the title is not entirely accurate I feel, as the study is more about using satellite retrievals to constrain the WRF-Chem aerosol representation. Perhaps the authors want to reconsider.

AC: This is an interesting suggestion. However, we prefer to keep the term "combining" instead of using the term "constraining". Indeed, our approach is fully based on the coupled use of numerical simulations and satellite retrievals, over a specific period and region. The term "constraining" might be wrongly interpreted, as a more global approach of the WRF-Chem aerosol representation.

RC2: Introduction, lines 33-35: worth noting is that southern Africa, which is the focus of this study, by itself produces one-third of the global annual carbon emissions from BB, according to the cited Werf paper.

AC: Thank you for this comment. It has been included in the revised version of the manuscript.

RC2: Intro, line 38: Costantino and Breon 2013 is not really the right reference for documenting the aerosol transport. The African Easterly Jet-South is not yet active in July. Knippertz et al. 2017 might be a better fit for this, or references within.

AC: As suggested, we replaced the Costantino and Breon (2013) reference with that of Knippertz et al. (2017).

RC2: Intro line 38-40: I am not sure the anticyclonic circulation responsible for long-range transport off of the continent is well established by July. Adebiyi and Zuidema 2016 suggests it isn't. And neither does Fig. 4. Fig. 4 does show an anticyclonic circulation in a couple of places but I don't see those affecting the regions selected for this study, shown in Fig. 2. Given that the authors have the model winds at their fingertips, perhaps they could say more about the circulations affecting their selected regions? It seems like the land domain might also be influenced by the west African monsoon? Do any of the cited papers discuss July? I am not sure they do.

AC: This is an interesting remark. According to the study of Gargstang et al. (1996), anticyclonic circulation reaches an 80 % daily occurrence in July over the southern Africa. Subsidence also controls the horizontal (and the vertical) transport of aerosols. This point is now clarified in the revised version of the manuscript.

RC2: introduction, 2nd paragraph, p.2: portions of this discussion feel dated, through the focus on the aerosol above clouds and neglect of the boundary layer BBA. Try to update.

AC: Thank you for this remark. We added the following sentence in the revised manuscript at line 60, page 2: "It is worth noting that the transport of BBA in the boundary layer has also been reported by recent airborne observations (Zuidema et al. 2018, Haslett et al. 2019, Denjean et al. 2020, Kacarab et al. 2020, Redemann et al. 2021, Zhang and Zuidema, 2021).

RC2: p.3: This is a nice literature overview. I'm confused why the Denjean, Taylor, Pistone papers aren't included here. It's also a very long paragraph, could it be split into two?

AC: Thanks, we split the third long paragraph into two: from line 62 to 79 and from line 80 to 102. We added the suggested papers at line 83, page 3 in the revised manuscript.

RC2: p. 3 line 93: CLARIFY and ORACLES references should be included here.

AC: We agree with the reviewer and we added the Redemann et al. (2021) reference for ORACLES and the Haywood et al. (2021) reference for CLARIFY at lines 93-94, page 3 in the revised manuscript.

RC2: p. 3 line 95: what does ANR stand for?

AC: ANR stands for the French National Research Agency. We replaced "ANR" by "French National Research Agency" at line 95, page 3 in the revised manuscript.

RC2: P.4 line 106: remove 'to perform'

AC: Thanks, we removed "to perform" at line 106, page 4 in the revised manuscript.

RC2: P. 4 lines 115-118: would be nice to see more documentation of this, is this based on the authors' own analysis?

AC: We added the Waquet et al. (2020) reference in the revised manuscript. This is based on the POLDER-3/AERO-AC data.

RC2: p. 7 fig 1: why include August and September? They are not used. I would suggest removing these panels.

AC: We included August and September 2008 because the July-August-September quarter is the heart of the dry season. We wanted to qualitatively assess the biomass burning emission inventory of Turquety et al. (2014) used in our study in terms of fire locations from the literature. Figure 1 thus provides a first general, and larger view of BBA emissions over our region of interest.

RC2: p. 11 lines 291-292: on what basis do the authors believe that desert dust emissions from north Africa may significantly contribute to the total aerosol load?

AC: Thank you for this question. Deaconu et al. (2019) clearly showed the presence of desert dust above clouds with mean Angstrom exponent values down to 0.4 and associated mean AOD of 0.2 over the northern part (between 0° and 5°N) of the Southeast Atlantic Ocean studied from May to October 2006 to 2009 (see their Fig. 2-ab). We removed "significantly" in the revised manuscript (line 291, page 11) which could be overstated.

RC2: p. 13, figure 4: is the ECMWF reanalysis the Interim analysis or ERA5? also, the wind vectors are very difficult to read. Please replot with fewer and thicker vectors. It also seems to me, based on Fig. 9, that the fields at 500 hPa could be removed from Figures 4-6 without loss.

AC: The ECMWF reanalysis is the ERA-Interim. This point is now clarified in the revised manuscript. According to the reviewer's suggestion, we replotted Fig. 4 with fewer and thicker vectors to get a better reading of the wind speed and direction (see below).

[Figure]

**Figure 4: Monthly averaged wind speed (m. s⁻¹) and direction at 850 hPa (left), 700 hPa (middle) and 500 hPa (right) from ECMWF reanalysis (top) and simulated with the WRF-Chem initial configuration (bottom) for July 2008.**

We kept the meteorological fields at 500 hPa (about 5.5 km altitude) because BBA are lifted up to an altitude of 6 km over land, on average for July 2008. Therefore, the meteorological fields at 500 hPa also seem important to assess as they may condition the BBA transport.

RC2: p. 14-15, lines 347-349: I cannot see winds capable of a westward aerosol transport in Fig. 4. I think the authors may be confusing the meteorology of September, which I suspect is what the cited papers focus on, with that of July.

AC: This is an interesting remark. Thanks to the replotting of the wind vectors in Fig. 4, we can now better see the north-westward aerosol transport with south-westerly/westerly winds.

RC2: p. 16 lines 384-386: given the finding that GOCART seems to raise too much dust, why not include an experiment in which its presence is reduced? (also 'dusts' -> 'dust')

AC: The reviewer is right. As explained in the revised manuscript, the average values of AOD are highly overestimated in the WRF-Chem initial configuration (mean $AOD_{mod,565} \geq 1.5$) compared to the POLDER-3/GRASP retrievals (mean $AOD_{obs,565} \simeq 1.0$) over the northern half of Africa in the Sahara and Sahel region. The GOCART AFWA desert dust emission module used in WRF-Chem (Jones et al. 2010, 2012), based on the Marticorena and Bergametti (1995) scheme, thus seems to raise too much mineral dust over the Sahara/Sahel area. According to the recommendations of Flaounas et al. (2016), who made an assessment of atmospheric dust modelling performance by WRF-Chem (version 3.6) against MODIS observations on arid and semi-arid regions around the Mediterranean, including North Africa, we have applied the adjustment coefficient of 0.5 on the desert dust emission surface fluxes in the GOCART AFWA scheme.

RC2: p. 17 line 404: did the Koffi papers evaluate July explicitly? There is a strong seasonal cycle to the aerosol vertical structure, see, e.g. Redemann 2021, ACP

AC: Koffi et al. (2012, 2016) studied the mean aerosol extinction profiles retrieved by CALIOP and modelled by 12 global models from the AeroCom project over the 2007-2009 period for 12 key worldwide regions, including South Africa (SAF). They especially separated the results into four period: March-April-May, June-July-August, September-October-November and December-January-February. We qualitatively compared the mean aerosol extinction coefficient simulated by WRF-Chem for July 2008 with the one observed by CALIOP during the June-July-August quarter over SAF. In their study, Koffi et al. (2012, 2016) also showed a strong seasonal cycle to the aerosol vertical structure.

RC2: p. 18 line 435: "on the opposite" -> "In contrast"

AC: Thanks, we took into account this correction in the revised manuscript.

RC2: p. 18 line 439: are the authors suggesting an aerosol invigoration effect on shallow clouds? This seems very unlikely to me and I see no reference. I would suggest just attributing the cloud parametrization, although it confuses me the parameterization cited (Lin) is a microphysical parameterization according to Table 1. Wouldn't the boundary layer scheme be the more likely cause?

AC: The reviewer is right. The planetary boundary layer scheme used in our study (YSU, Hong et al. 2006) would be the more likely cause. We modified the sentence at lines 437-440, page 18, as follows: "The difference between

the model and the POLDER-3 retrievals could come from the planetary boundary layer scheme (YSU, Hong et al. 2006) used in the WRF-Chem reference configuration."

RC2: p. 18 line 464: sea salt mixed with smoke I would think. What does the model say?

AC: According to the model, the aerosol layer located between 0 and 0.8 km corresponds to sea salt aerosols (purely scattering coarse mode aerosol) located in the boundary marine layer (Deaconu et al. 2019; Peers et al, 2019). We modified the sentence at lines 462-464, page 19, as follows: "For July 12, 2008, the aerosol layer located between 0 and 0.8 km above sea level, generally detected by CALIOP and simulated with the WRF-Chem partially constrained configuration, corresponds to sea salt aerosols in the model, as usually observed in the boundary marine layer (Deaconu et al. 2019; Peers et al, 2019)."

RC2: P. 18 lines 467-468: see also Shinozuka 2020 ACP, which shows many models share these aerosol layer altitude biases.

AC: Indeed, this is an excellent study that we could build on in a future study. The suggested reference has been added in the revised manuscript.

RC2: p. 19, fig. 10 left panel+discussion: is this for the free-tropospheric aerosol layer? An average over the full column? I'm confused by this, and how the coarse mode is increasing with distance. Incidentally the orange star is placed near Ascension Island, and some assessment could be done using the LASIC datasets if interested.

AC: The aerosol volume size distributions are vertically integrated over the whole atmospheric column. We added "whole atmospheric column" in parenthesis after "vertically integrated" at line 474, page 19, and at line 478, page 20 in the revised manuscript. We concluded that the increase in the coarse mode during the aerosol transport, especially during the long-range transport, is due to the more pronounced presence of sea salt and desert dust aerosols (see Fig. A at the beginning). The reviewer is right that some assessment could be done using the LASIC datasets but only a qualitative comparison due to the non-matching period of study.

RC2: p. 22 lines 532-534: no mention of dust in the DACCIWA data description, further leading me to think an experiment should be done where it is removed from WRF-Chem and a further assessment done to see what additional changes have to be incorporated, for WRF-Chem to match the satellite retrievals.

AC: The reviewer is right. We only compared the main aerosol compounds of biomass burning with those measured during the DACCIWA airborne campaign. We conducted three numerical experiments with WRF-Chem to assess the possible influence of desert dust aerosols over the Southeast Atlantic region and their effects on the spectral dependence of SSA from aerosols, as illustrated in Fig. B (see below). The first scenario corresponds to the initial GOCART AFWA desert dust aerosol emissions scheme (DUST x1, initial configuration in orange curves). The second corresponds to the modified GOCART AFWA desert dust aerosol emissions scheme (DUST x0.5, partially constrained configuration in blue curves). The last scenario corresponds to the removal of desert dust aerosol in WRF-Chem (no DUST, yellow curves).

[Figure]

**Figure B: Spectral dependencies of the single scattering albedo of aerosols simulated by three WRF-Chem experiments (colored curves) over land in clear sky (12°E-30°E, 10°S-0°S, dashed curves) and above clouds of the Southeastern Atlantic Ocean (0°E-15°E, 15°S-5°S, dashed curves), and retrieved by POLDER-3 (GRASP in dashed black curve, AERO-AC in solid black curve) on average for July 2008. The black vertical error bars correspond to the uncertainties associated with the POLDER-3 data with an accuracy of ±0.05 (per pixel estimates) over the whole spectral domain. The first WRF-Chem scenario corresponds to the initial GOCART AFWA desert dust aerosol emission scheme (DUST x1, initial configuration, orange curves). The second WRF-Chem scenario corresponds to the modified GOCART AFWA desert dust aerosol emission scheme (DUST x0.5, partially constrained configuration, blue curves). The last scenario of WRF-Chem corresponds to the removal of desert dust aerosols in the model (no DUST, yellow curves).**

Figure B compares the spectral values of SSA resulting from these three WRF-Chem experiments (colored curves) with the retrievals from POLDER-3/GRASP (solid black curve) over land in clear sky (12°E-30°E, 10°S-0°S) and from POLDER-3/AERO-AC (dashed black curve) above clouds over the SEAO (0°E-15°E, 15°S-5°S), on average for July 2008. The three sets of simulations show that the effect of desert dust aerosols is slightly visible in the near infrared (1,000 nm) with a small decrease in absorption with increasing desert dust aerosols concentration: the SSA values increase of approximately 0.005 between the desert dust aerosols-free scenario (no DUST, yellow curves) and the WRF-Chem partially constrained configuration (DUST x0.5, blue curves), both in clear and cloudy atmospheres. In the ultraviolet (300-400 nm), the high absorption of desert dust aerosols is absent: the SSA spectral values only decrease by about 0.001 and 0.003 between the WRF-Chem partially constrained configuration (DUST x0.5, blue curves) and the desert dust aerosols-free scenario (no DUST, yellow curves), respectively over land in clear sky and above clouds over the SEAO. We can conclude from these WRF-Chem experiments that the presence of desert dust aerosols area has little influence on the SSA modeling in our study. Therefore, we kept desert dust aerosols to obtain simulated aerosol chemical composition and optical properties the more realistic as possible.

RC2: p. 25 line 568: "an increased of the aerosol absorption' => 'increased aerosol absorption'

AC: Thanks, we took into account this correction in the revised manuscript.

RC2: p. 25 line 573: "consistently' ->'consistent' here and elsewhere.

AC: Thanks, we took into account your corrections at line 403, page 17, lines 562 and 572, page 25 and line 668, page 29 in the revised manuscript.

RC2: p. 28, line 643: is there any evidence for more desert dust becoming incorporated into the aerosol layer as it moves westward other than from the WRF-Chem model? Any observations of this?

AC: Thank you for this question. Please see Fig. A at the beginning and its associated discussion, and the Deaconu et al. (2019) paper.

RC2: p. 29 line 661: remove "that", 'Located" "the geographic coordinates area"

AC: Thanks, we took into account your corrections. Besides, we removed "of geographic coordinates" everywhere in the revised manuscript.

RC2: p. 30 line 692: is the Leahy 2007 representative of July? Eck 2013, Zuidema 2018 both show a strong seasonal evolution to the SSA so good to make sure it's about the same time frame.

AC: Thank you for this question. The mean SSA of 0.85±0.02 (550 nm) obtained during the SAFARI-2000 airborne campaign is not representative of July (Leahy et al. 2007). We modified the sentence at line 691, page 30, as follows: "In particular, the simulated mean SSA is consistent with the mean value of 0.85±0.02 observed during SAFARI-2000, although not fully representative of the same period of the BB season (Leahy et al. 2007)."

For instance, Peers et al. (2016) also showed a strong seasonal evolution of the SSA over the SEAO with a higher aerosol absorption in July (see their Fig. 1).

RC2: p. 33 references: the formatting of the references is not consistent, check to make sure they fit the ACP format. It also seems like there are more references than are used?

AC: Thank you for this information. We used Zotero to automatically create the bibliography.

RC2: Additional reference suggestions:

Adebiyi, A. and P. Zuidema, 2016: The role of the southern African easterly jet in modifying the southeast Atlantic aerosol and cloud environments. Q. J. R. Meteorol. Soc., 142, p. 1574-1589 doi: [10.1002/qj.2765]

Haywood, J. M., S. Abel, P. Barrett, et al, 2020: Overview: The CLoud-Aerosol-Radiation Interaction and Forcing: Year-2017 (CLARIFY-2017) measurement campaign, Atmos. Chem. Phys., 21, p. 1049-1084, doi:10.5194/acp-21-1049-2021

Kacarab, M., et al, 2020: Biomass Burning Aerosol as a Modulator of Droplet Number in the Southeast Atlantic Region. Atmos. Chem. Phys., 20, p. 3029-3040, doi:10.5194/acp-20-3029-2020 - this focuses on the month of August, so slightly later than July, nevertheless supports the idea that in the early part of the BB season, much aerosol is also present in the boundary layer, and, indicates the microphysical implication.

Knippertz, P., Fink, A. H., Deroubaix, A., et al..: A meteorological and chemical overview of the DACCIWA field campaign in West Africa in June–July 2016, Atmos. Chem. Phys., 17, 10893–10918, https://doi.org/10.5194/acp-17-10893-2017, 2017

Redemann, J., R. Wood, P. Zuidema, et al, 2020: An overview of the ORACLES (ObseRvations of Aerosols above CLouds and their intEractionS) project: aerosol-cloud-radiation interactions in the Southeast Atlantic basin. Atmos. Chem. Phys., 21, p. 1507-1563, doi:10.5194/acp-21-1507-2021

Shinozuka, Y., P. E. Saide, G. A. Ferrada, et al. 2020: Modeling the smoky troposphere of the southeast Atlantic: a comparison to ORACLES airborne observations from September of 2016. Atmos. Chem. Phys., 20, p.11,491-11,526, doi:10.5194/acp-20-11491-2020 - could be interesting (in a future study) to see how the new WRF-Chem representation does within this same comparison, the data are accessible in an easy meta format. Also useful for more context on where WRF-Chem falls in the pantheon of aerosol model representations. The downward 'slumping' of the aerosol layer shown in Fig. 9 is also shown to be common to many other aerosol modelling efforts.

J. Zhang and P. Zuidema, 2021: Sunlight-absorbing aerosol amplifies the seasonal cycle in low cloud fraction over the southeast Atlantic: Atmos. Chem. Phys. Disc., doi:10.5194/acp-2021-275 - more information on July aerosol observations.

AC: References:

[revised manuscript text omitted]

RC3: Review of: Combining POLDER-3 satellite observations and WRF-Chem numerical simulations to derive biomass-burins aerosol properties over the Southeast Atlantic region. Authors: A. Simeon, F. Waquet, J-C Pere, F. Ducos, F. Thieuleux, F. Peers, S. Turquety, and I. Chiapello.

RC3: Recommendation: Accept with Minor Revisions

RC3: In this study, the authors focus on biomass burning (BB) particle plumes transported above clouds over the Southeast Atlantic (SEA) region, off the southwest coast of Africa. They employ simulations from a regional model (WRF-Chem) coupled with meteorological reanalyzes data and aerosol retrievals from POLDER in clear sky (POLDER/GRASP) and cloudy scenes (POLDER-3/AERO-AC), to better characterize the physico-chemical and absorption properties of aerosols.

RC3: Other reviewers have asked relevant questions, so I will just add some additional comments below:

RC3: Page 6, line 155: What is the temporal resolution of the WRF-Chem configuration?

AC: Thank you for this question. The temporal resolution of the WRF-Chem configuration is hourly. We added the following sentence at line 155, page 6: "The temporal resolution of the WRF-Chem numerical simulations is hourly.". This point is now clarified in the revised manuscript.

Are the vertical levels evenly distributed from surface to 50hPa or the vertical resolution is finer near the surface? Could you clarify in the text?

AC: Thank you, this point is now clarified in the revised manuscript. We adopted the hybrid sigma-pressure vertical coordinate which is terrain-following near the surface with a finer resolution than at the fixed top pressure (flat). We modified the sentence at lines 155-156, page 6, as follows: "The atmospheric layer is divided into 50 vertical levels using the hybrid sigma-pressure vertical coordinate. The vertical levels are terrain-following near the surface with a finer resolution than at the upper pressure level set at 50 hPa.".

Line 157: Whilst I agree that the first half of July is representative for the whole month, this short period is not representative for the entire biomass-burning season. As Adebiyi et al, 2015 and Deaconu et al, 2019 showed, September and October months are characterized by different meteorological conditions and larger amounts of BBA transported over the SEAO. Also, the BBA are lifted at higher altitudes, limiting the contact aerosol-clouds. How do you justify choosing this period to study?

AC: This is an interesting remark. We chose to study this period because the aerosol absorption is the strongest (very low SSA). If we can go low enough for this period, our approach is more likely to be able (in theory) to reproduce the higher SSA observed in August and September. The SSA cycle is shown in Fig. 1 in Peers et al. (2016) for the fire season 2006 over the SEAO (5°N-30°S, 20°E-20°W). Furthermore, Figure A (not shown in the article) shows the monthly variabilities of spectral ACAOD (left) and ACSSA (right) retrieved by POLDER-3/AERO-AC above clouds in its native resolution (6 x 6 km$^2$), during the fire season 2008, and averaged over the Southeast Atlantic (SEA) region (20°W-30°E, 39.9°S-10°N).

[Figure]

POLDER-3/AERO-AC retrievals: fire season 2008

| ●—490 nm | ◆—550 nm | ▲—670 nm | ■—865 nm |

**Figure A: Monthly variabilities of aerosol optical depth (ACAOD, left) and single scattering albedo (ACSSA, right) retrieved by POLDER-3/AERO-AC above clouds at a 6 x 6 km² spatial resolution from blue to near infrared over the Southeast Atlantic region (20°W-30°E, 39.9°S-10°N) during the fire season 2008.**

Figure A indicates that the lowest values of ACSSA, corresponding to the strongest absorptions, are obtained for the month of July 2008. For example, the value of POLDER-3/AERO-AC ACSSA at 865 nm (red curve, right) decreases from 0.837 in June to 0.825 in July and then increases to 0.887 in October.

What was the computational cost of the 30 days (with 15 days spin-up) simulation and could you apply the optimized model configuration over September/October 2008 to check the consistency over this period?

AC: The computational cost of the 30 days (with 15 days spin-up) simulation was less than 1 week. It depended on the number of available and used processors. As suggested, we applied the WRF-Chem optimized configuration until the end of September 2008 to test its ability to represent the absorption cycle observed during the fire season of 2008. Figure B shows the temporal evolutions of the ACAOD (left) and ACSSA (right) retrieved by POLDER-3/AERO-AC (solid line curves) and simulated with the WRF-Chem optimized configuration (dotted curves) above clouds at 490, 550 and 865 nm (blue, green and red curves, respectively), during the July-August-September 2008 quarter.

[Figure]

**Figure B: Temporal evolutions of the ACAOD (left) and the ACSSA (right) retrieved by POLDER-3/AERO-AC (solid line curves) and simulated with the WRF-Chem optimized configuration (BCx2, OC/2.5 with 2.5% of BrOC, dotted curves) above clouds over the Southeast Atlantic Ocean at 490, 550 and 865 nm (blue, green and red curves, respectively). The optical properties are averaged geographically within the coordinate study area of (0°E-15°E, 15°S-5°S) and monthly over the July-September 2008 period. The vertical error bars correspond to the uncertainties associated with POLDER-3 data with an accuracy of ±20 % on the aerosol optical depth and ±0.05 on the single scattering albedo over the entire spectral domain.**

On the one hand, Figure B shows that the WRF-Chem optimized configuration reproduces in part at least the increase in ACAOD retrieved by POLDER-3/AERO-AC between July and August, but does not reproduce the variability observed between August and September 2008. For September 2008, the WRF-Chem optimized configuration fails to adequately simulate the aerosol concentration peak which is typically observed at this time of year over the Southeast Atlantic region. Indeed, the WRF-Chem optimized configuration underestimates the amounts of aerosols transported over the marine stratocumulus clouds with an average bias of -0.24 for ACAOD at 550 nm compared to the POLDER-3/AERO-AC retrievals in September 2008. This underestimation could be related to uncertainties about the evolution of biomass burning emissions during the season in the inventory used (APIFLAMEv1, Turquety et al. 2014). Recently, a numerical sensitivity study was conducted by Pan et al. (2019) by testing six global biomass burning emissions inventories (excluding APIFLAMEv1) in the NASA's GEOS-Chem chemistry-transport model. In particular, this modelling study focused on the South African continent in September 2008. The results of the study showed an underestimation of the simulated AODs over land for these six inventories, with a maximum bias of 0.23 compared to MODIS-Aqua data at 550 nm and at least 50% compared to data from the AERONET site in Mongu, Zambia. This discrepancy between modelling and observations could not be explained by different synoptic conditions because the meteorology and aerosol mechanisms (transport and removal) were identical for these six numerical experiments. The authors of this study therefore suggest that estimates of the amounts of aerosols emitted by biomass burning should be revised high, in particular by updating the emission factors according to the season and the conditions of activity of the fires.

On the other hand, Figure B shows that the WRF-Chem optimized configuration struggles to represent very satisfyingly the seasonal cycle of ACSSA retrieved by POLDER-3/AERO-AC, which is characterized by a decrease in aerosol absorption (increase in ACSSA values) during the advance of the fire season. This result suggests that the chemical composition of the biomass burning aerosol plumes, especially the carbonaceous aerosols, changes during the dry season, likely in relation to a change in the type of fuel burned and in combustion conditions (Zuidema et al. 2016b; Pan et al. 2019). It is therefore necessary to study the aerosols physico-chemical

and optical properties on a month-by-month basis in order to better understand and constrain the biomass burning aerosols absorption cycle over the South-East Atlantic region in WRF-Chem.

RC3: Figures 1 and 2: From these figures looks like you could have chosen a different study area for the clear-sky cases, say between 5° and 15° N, that would have covered more of the biomass-burning emissions (correlated with the PM2.5 in Fig.1) and also more clear-sky days (Fig.2). Why did you choose this particular box?

AC: This is an interesting question. We chose this particular box (12°E-30°E, 10°S-0°S) because the aerosol load is maximum (see Fig. 7, left, in the revised manuscript), and the uncertainties related to the retrievals of the aerosol optical properties by POLDER-3/GRASP are the lowest.

RC3: Page 10, Fig2b: You are showing number of observation days of clear and cloudy scenes. It would be useful to plot also the data with coincident aerosol retrievals (e.g., number of observation days used in the study) for clear and cloudy skies.

AC: Thank you for this suggestion. In fact, an inversion is performed for each of these days.

RC3: Page 16, line 395: What is the scale of the biases between the AOD in clear-sky POLDER/GRASP compared to WRF-Chem reference configuration? From Fig. 7, it looks like the model is still strongly underestimating the AOD over land. Since the range of uncertainties for BB emission inventories reported in literature in 2 to 4, why did you choose to scale the APlFLAMEv1 with a factor of 1.5?

AC: This is an interesting question. The scale of the biases between the AOD in clear-sky POLDER/GRASP compared to WRF-Chem reference configuration is 2.24. We first applied a multiplying factor of 2 to the APIFLAMEv1 BB emissions inventory and the result was not concluding. Over land (12°E-30°E, 10°S-0°S), simulated AOD values ($AOD_{mod,565} = 0.98$) were in much better agreement with those retrieved by POLDER-3/GRASP ($AOD_{obs,565} = 1.10$) with a moderate underestimation of 0.12 on a geographical average. Nevertheless, above the ocean, simulated AOD values ($AOD_{mod,565} = 0.86$) were overestimated compared to POLDER-3/GRASP retrievals ($AOD_{obs,565} = 0.61$), with a positive bias of 0.25 on a geographical average. This could be explained by a higher emitted and then transported aerosol load. This could also be interpreted as a numerical diffusion problem of a dense biomass burning aerosol plume, which causes a spread of the load between the continent and the ocean due to the width of the mesh of WRF-Chem (as a reminder, a grid of 30x30 km$^2$). Finally, the adjustment factor of 1.5 was chosen as being a good compromise, even if small biases were still present. This point is now clarified in the revised manuscript.

RC3: Page 17, line 408: Deaconu et al., 2019 showed that oxygen pressure method underestimates the cloud height compared to the CALIOP retrieval, by about 2-300 m for low clouds. Therefore, the WRF-Chem underestimation of cloud top could not be as high as 500 m. Could you have used the CTH from CALIOP instead of POLDER for the model optimization (or scale up the POLDER CTH using CALIOP retrievals)?

AC: This is an interesting suggestion. We applied the empirical relationship proposed by Deaconu et al. (2019) to correct the altitude of the cloud top height retrieved by POLDER-3. The advantage of the corrected POLDER-3 data is the much greater spatial coverage than that of the CALIOP lidar, which allows to obtain better statistics. This point is now clarified in the revised manuscript.

RC3: In the beginning of the paper, you mention having simulated only half of July as representative for the entire month. In Fig.2 you mention '01-15 July period' and everywhere else you mention only 'July 2008', which leaves the wrong impression the data (satellite and/or model) are averaged over the entire month. Please clarify in the captions and in the text where necessary.

AC: This point has been now clarified in the revised manuscript. The simulation of the first half of July 2008 only concerns the numerical experiments. The assessments of the WRF-Chem initial and reference configurations were done for the whole month. We modified the text at lines 157-160, page 6, as follows: "We have simulated the first half of July 2008 (plus 15 days spin-up) for the numerical experiments to reduce numerical costs. The computation time was about three days for this time period and for one numerical simulation against about six days for the whole month (plus 15 days spin-up). Furthermore, the first half of July 2008 appears to be representative of July 2008. Indeed, the Above-Cloud Aerosol Optical Depth (ACAOD) and the Above-Cloud Single Scattering Albedo (ACSSA) retrieved by POLDER-3/AERO-AC at 550 nm are respectively 0.45 and 0.85 over the first half of July 2008 and 0.46 and 0.85 over July 2008 on average over our combined studied areas (black frames in Figure 2).". We added the following sentence at line 304, page 11: "It is worth noting that the assessment conducted in the first step is done for the whole month of July 2008.".

RC3: Minor corrections:

RC3: Page 10, line 279: '…to evaluate the aerosol extinction simulated with WRF-Chem at 550, as well as…'

RC3: Page 16, line 360: '…This bias could be due to…'

RC3: Page 17, line 399: '…besides the simulated aerosol loads…'

RC3: Page 18, line 431: '…WRF-Chem configuration simulates well the …'

line 447: '…depolarization ratio method…'

RC3: Sometimes the phrases are too long, and there are missing commas or linking words that could make reading more elegant and easier.

AC: We appreciate the suggested corrections and took them into account. We also tried to reduce the sentences as much as possible.

AC: References:

Deaconu, Lucia T., Nicolas Ferlay, Fabien Waquet, Fanny Peers, François Thieuleux, and Philippe Goloub. 2019. « Satellite Inference of Water Vapour and Above-Cloud Aerosol Combined Effect on Radiative Budget and Cloud-Top Processes in the Southeastern Atlantic Ocean ». Atmospheric Chemistry and Physics 19 (17): 11613-34. https://doi.org/10.5194/acp-19-11613-2019.

Pan, Xiaohua, Charles Ichoku, Mian Chin, Huisheng Bian, Anton Darmenov, Peter Colarco, Luke Ellison, et al. 2019. 'Six Global Biomass Burning Emission Datasets: Inter-Comparison and Application in One Global Aerosol Model'. Preprint. Aerosols/Atmospheric Modelling/Troposphere/Physics (physical properties and processes). https://doi.org/10.5194/acp-2019-475.

Peers, F., N. Bellouin, F. Waquet, F. Ducos, P. Goloub, J. Mollard, G. Myhre, et al. 2016. 'Comparison of Aerosol Optical Properties above Clouds between POLDER and AeroCom Models over the South East Atlantic Ocean during the Fire Season'. Geophysical Research Letters 43 (8): 3991–4000. https://doi.org/10.1002/2016GL068222.

Turquety, S, Laurent Menut, B Bessagnet, Alessandro Anav, Nicolas Viovy, Fabienne Maignan, and M Wooster. 2014. APIFLAME v1.0: High-Resolution Fire Emission Model and Application to the Euro-Mediterranean Region. Vol. 7. https://doi.org/10.5194/gmd-7-587-2014.

Zuidema, Paquita, Jens Redemann, James Haywood, Robert Wood, Stuart Piketh, Martin Hipondoka, and Paola Formenti. 2016b. 'Smoke and Clouds above the Southeast Atlantic: Upcoming Field Campaigns Probe Absorbing Aerosol's Impact on Climate'. Bulletin of the American Meteorological Society 97 (7): 1131–35. https://doi.org/10.1175/BAMS-D-15-00082.1.

EC: This is a useful study, it's great to see the WRF-Chem interrogated to such an extent. The authors sufficiently addressed the reviewers' comments that I am not inclined to send the manuscript back to them, but I do have some comments I'd like to see addressed before this goes to publication. The following statements refer to the 'tracked changes' document:

EC: Line 41: the statement that anticyclonic conditions occur 80% of the time in July needs more specificity. I see the authors mentioning the Swap and Garstang papers in the response to the reviewers as the source for this information, I'm not sure how those authors described the large-scale flow, but if you are going to go with this, at least include the citation and describe what they say with more detail. But keep in mind that July is the January of the southern hemisphere and such a land-based circulation is not likely to extend very far offshore. I don't see much of an anticyclonic circulation present in the figure 4. Zhang&Zuidema 2021 show above-cloud winds are weak at Ascension in July. Your nice fig. 7 also doesn't show evidence of a land-based anticyclonic circulation, the aerosol is just zonally diffused away from the continent. What would be more straightforward is to just say that there are prevailing if weak easterlies from 850-700hpa that advect aerosol westward. Your figures support that and then you'd be done.

AC: Thanks for this suggestion, we agree with your analysis. The text has been simplified in the revised 'tracked changes' manuscript. The paragraph from lines 40 to 44, page 2:

"This westward long-distance transport is favoured by the predominant anticyclonic conditions over the southern Africa during the dry season. It is worth noting that anticyclonic circulation reaches an 80 % daily occurrence in July over the southern Africa. Subsidence also controls the horizontal and the vertical transport of aerosols (Cahoon et al., 1992; Garstang et al., 1996; Swap et al., 1996)."

has been replaced by the sentence

"This westward aerosol transport is favoured by prevailing easterlies winds at 850 hPa and 700 hPa (Zuidema et al., 2018).".

EC: Line 250 in tracked changes: 'against about'? Do you just mean 'about'?

AC: Thank you, this point is now clarified in the revised 'tracked changes' manuscript. We modified the sentence at lines 250-251, page 9, as follows:

"The computation time for a numerical simulation was about three days for this time period and about six days for the whole month (plus 15 days spin-up).".

EC: Fig4: wind vectors difficult to see in the top 2 rows, can those be redone to be similar to the bottom two rows?

AC: Thank you for this suggestion. The length of the wind vectors is proportional to the wind speed. This is why the wind vectors are smaller at the top of the figures, corresponding to low wind speed areas (in blue), than at the bottom, corresponding to high wind speed areas (in red). We increased the width of the arrows and we modified their number in order to get the best compromise (see below).

[Figure]

**Figure 4: Monthly averaged wind speed (m. s$^{-1}$) and direction at 850 hPa (left), 700 hPa (middle) and 500 hPa (right) from ECMWF ERA-Interim reanalysis (top) and simulated with the WRF-Chem initial configuration (bottom) for July 2008. Current figure in the tracked changes manuscript.**

[Figure]

**Figure 4: Monthly averaged wind speed (m. s$^{-1}$) and direction at 850 hPa (left), 700 hPa (middle) and 500 hPa (right) from ECMWF ERA-Interim reanalysis (top) and simulated with the WRF-Chem initial configuration (bottom) for July 2008. New figure in the revised tracked changes manuscript.**

EC: P.21 last 2 lines: this statement is simply incorrect. There is no anticyclonic circulation, at least not one over land, visible in fig. 4. This is the one I believe the cited papers would be referring to. The altitude of the smoke emission is low enough, especially in the model, that the anticyclonic circulation over the ocean, around the south Atlantic sea level pressure high, would have more of an influence in distributing the aerosol over the ocean than any land circulation. Again, I think you can just say here that the winds blow westward off of land north of 20S.

AC: Thank you for this correction. This point is now clarified in the revised 'tracked changes' manuscript. We modified the text at lines 507-510, pages 21-22, as follows:

"North of the 20[th] parallel south, the mean simulated winds blow westward off of land and favour a westward transport of BBA plumes emitted from the Southern African continent.".

EC: P.25: an injection height of 6km seems high to me, your own caliop data shows the aerosol only extends up to 4km. But the aerosol advected over the ocean in the model is clearly too low. Can the authors comment? Does an injection height of 6km still mean that a lot of the aerosol is placed lower?

AC: Indeed, the text p. 25 indicates that the injection height of BBA in the model is up to about 6 km altitude over land, so this can be considered as a maximum height of injection. Such a high injection height is supported by previously published CALIOP observations analyzed over the Southern African continent (Koffi et al., 2012, 2016).

Figure A illustrates that the BBA simulated with the WRF-Chem PC configuration, when emitted from different source areas over land, are mostly located below 4-5 km altitude although injected into the atmosphere up to about 6 km altitude (on average for July 2008).

[Figure]

**Figure A: Left, monthly averaged PM$_{10}$ mass concentrations emitted at surface (in mg. m$^{-3}$) simulated with the WRF-Chem partially constrained configuration for July 2008. Right, vertical profiles of PM$_{10}$ mass concentrations (in mg. m$^{-3}$) injected into the atmosphere over three African biomass burning major sources simulated by the WRF-Chem partially constrained configuration, on average for the same time period.**

Figure 9 shows two examples of aerosol vertical profiles, but it should be kept in mind that they correspond to BBA events that are already transported off land, over the Atlantic Ocean. The altitude of BBA emitted from African fire source areas and then transported westward is known to decrease with distance from the sources (e.g., Deaconu et al., 2019).

In summary, it is likely that a large part of the BBA is emitted lower in altitude than the maximum injection height of 6 km altitude in the model, as shown by Figure A. Moreover, in a consistent way, Fig. 9 shows, at least for 2 examples of vertical profiles, that the aerosol plumes advected over the ocean tend to be too low in altitude compared to CALIOP observations.

We modified in the revised 'tracked changes' manuscript the sentence at lines 691-692, page 30:

"It is worth noting that many models share these aerosol layer altitude biases over this region (Shinozuka et al., 2020)"

by

"It is worth noting that many models share these aerosol layer altitude biases over this region (Shinozuka et al., 2020) and that the BBA altitude may decrease during the transport (Deaconu et al., 2019).".

EC: P. 28, fig. 9: this is a nice figure. We know that there is plenty of smoke in the boundary layer in July, from the Ascension Island measurements. CALIOP won't be able to discern this, but WRF-Chem might actually be doing better in the boundary layer than this figure would suggest. Would it be possible to show the WRF-Chem extinction from smoke or BC alone? Or alternatively could the authors mention the PC boundary layer smoke mass concentrations somewhere and place them in context with the Ascension Island rBC and, if desired, ACMS measurements of OA?

AC: Thank you for these comments and suggestions. Unfortunately, it is not possible to show the WRF-Chem vertical profiles of extinction for smoke or BC alone. The WRF-Chem model calculates the extinction for each chemical species but the output variable is a weighted average (over volume) of each species. Besides this limitation, additional answers to your questions are brought just hereafter (Figure B).

EC: P. 31, line 715: presumably more of the long-range transport aerosol is in the boundary layer in the model, so the vertically-integrated volume size distribution will contain a larger contribution from the BL. This would suggest the coarse mode aerosol is mostly or entirely sea spray aerosol. Have the authors examined the AERONET data from Ascension? I also include micropulse lidar derived profiles of extinction and the depolarization ratio for July 2016 and 2017, these suggest some dust but it's not much (extinction retrieved similarly to the AOD constrained approach of MPLNET; https://agupubs.onlinelibrary.wiley.com/doi/10.1029/2018JD028867). It seems unlikely to me that there is more mass in dust than in sea spray, as stated by the authors on p. 34.

[Figure]

AC: Thank you for this remark. We already looked at the AERONET data from Ascension Island but unfortunately there were no available aerosol data for the whole month of July 2008. Figure B illustrates the monthly averaged vertical profiles of the aerosol extinction coefficient at 550 nm (in $km^{-1}$, red dotted curve) and of the volume concentrations (in $\mu g. \, m^{-3}$) of carbonaceous aerosols, i.e., black carbon (BC, black curve) and organic carbon (OC, brown curve), and sea salt aerosols (SEAS, blue curve) simulated with the WRF-Chem PC configuration for a pixel (14.93°W, 4.00°S) representative of the long-range transport of the BBA plume, for July 2008.

[Figure]

**Figure B: Monthly averaged vertical profiles (left) of the aerosol extinction coefficient at 550 nm (in km$^{-1}$, red dotted curve) and of the volume concentrations (in µg. m$^{-3}$) of carbonaceous aerosols, i.e., black carbon (BC, black curve) and organic carbon (OC, brown curve), and sea salt aerosols (SEAS, blue curve) simulated with the WRF-Chem PC configuration for a pixel (orange star, (14.93°W, 4.00°S)) representative of the long-range transport of BBA plume, indicated on the map of the total aerosol optical thickness at 565 nm in clear sky (right), for July 2008. The blue (14.83°E, 4.00°S) and red (4.82°E, 4.00°S) stars symbolize the emission sources of BBA over the Southern African continent and the transport of the BBA plume over the Southeast Atlantic Ocean, respectively.**

Figure B shows that the volume concentration of SEAS is largely greater than that of BC in the boundary layer in the area of Ascension Island on average for July 2008. This suggest that the coarse mode aerosol is mostly sea spray aerosols in the boundary layer, although desert dust aerosols may be located higher in altitude, thus contributing to the vertically-integrated volume size distribution of aerosols shown in Fig. 10.

Accordingly, the sentence at line 715, p.31, has been slightly modified to clarify this point: "During the aerosol transport, there is a gradual appearance of larger particles (diameter > 1 µm), which may be linked to aging processes and also to a slightly more pronounced influence of other larger particles, such as sea salt aerosols **mostly in the boundary layer** and **possibly** desert dust aerosols **at higher altitudes**, that may come across the BBA plumes over the SEAO.".

EC: Fig. 12: doesn't seem consistent with ACSM values at Ascension. BC low compared to Z&Z2021.

AC: Thank you for this comment. In fact, our studied areas, i.e., over land (12°E-30°E, 10°S-0°S) and above stratocumulus marine clouds (0°E-15°E, 15°S-5°S), and our time period in Fig. 12 are different. It would be tricky to directly compare our simulated BC mass concentrations, that are vertically integrated (as shown in Fig. 12) to those measured near surface at Ascension Island. We completed the sentence at lines 772-773, page 35:

"However, we can note that BC contributions simulated with the WRF-Chem PC configuration (≤ 2.1 %) in our studied areas are smaller than those measured during DACCIWA and LASIC (Zuidema et al., 2018b; Zhang and Zuidema, 2021) campaigns.".

EC: P. 46 line 944: ORACLES->LASIC

AC: Thank you for this suggestion. We took into account this change in the revised 'tracked changes' manuscript.

EC: P. 50 line 1027: LASIC could also be mentioned here. Some of its data could have been used for this study, although what the authors have done is already interesting.

AC: We agree with the Editor and we also mentioned LASIC in the revised 'tracked changes' manuscript at line 1027, page 50. Furthermore, we added the LASIC reference (Zuidema et al., 2018a) in the introduction at line 177, page 6.

AC: References

Deaconu, L. T., Ferlay, N., Waquet, F., Peers, F., Thieuleux, F., and Goloub, P.: Satellite inference of water vapour and above-cloud aerosol combined effect on radiative budget and cloud-top processes in the southeastern Atlantic Ocean, Atmospheric Chem. Phys., 19, 11613–11634, https://doi.org/10.5194/acp-19-11613-2019, 2019.

Koffi, B., Schulz, M., Bréon, F.-M., Griesfeller, J., Winker, D., Balkanski, Y., Bauer, S., Berntsen, T., Chin, M., Collins, W. D., Dentener, F., Diehl, T., Easter, R., Ghan, S., Ginoux, P., Gong, S., Horowitz, L. W., Iversen, T., Kirkevåg, A., Koch, D., Krol, M., Myhre, G., Stier, P., and Takemura, T.: Application of the CALIOP layer product to evaluate the vertical distribution of aerosols estimated by global models: AeroCom phase I results, J. Geophys. Res. Atmospheres, 117, https://doi.org/10.1029/2011JD016858, 2012.

Koffi, B., Schulz, M., Bréon, F.-M., Dentener, F., Steensen, B. M., Griesfeller, J., Winker, D., Balkanski, Y., Bauer, S. E., Bellouin, N., Berntsen, T., Bian, H., Chin, M., Diehl, T., Easter, R., Ghan, S., Hauglustaine, D. A., Iversen, T., Kirkevåg, A., Liu, X., Lohmann, U., Myhre, G., Rasch, P., Seland, Ø., Skeie, R. B., Steenrod, S. D., Stier, P., Tackett, J., Takemura, T., Tsigaridis, K., Vuolo, M. R., Yoon, J., and Zhang, K.: Evaluation of the aerosol vertical distribution in global aerosol models through comparison against CALIOP measurements: AeroCom phase II results, J. Geophys. Res. Atmospheres, 121, 7254–7283, https://doi.org/10.1002/2015JD024639, 2016.

Zhang, J. and Zuidema, P.: Sunlight-absorbing aerosol amplifies the seasonal cycle in low-cloud fraction over the southeast Atlantic, Atmospheric Chem. Phys., 21, 11179–11199, https://doi.org/10.5194/acp-21-11179-2021, 2021.

Zuidema, P., Alvarado, M., Chiu, C., DeSzoeke, S., Fairall, C., Feingold, G., Freedman, A., Ghan, S., Haywood, J., and Kollias, P.: Layered Atlantic Smoke Interactions with Clouds (LASIC) Field Campaign Report, edited by: Stafford, R, DOE/SC-ARM-18-018, ARM Climate Research Facility, Report DOE/SC-ARM-18-018 …, 2018a.

Zuidema, P., Sedlacek, A. J., Flynn, C., Springston, S., Delgadillo, R., Zhang, J., Aiken, A. C., Koontz, A., and Muradyan, P.: The Ascension Island Boundary Layer in the Remote Southeast Atlantic is Often Smoky, Geophys. Res. Lett., 45, 4456–4465, https://doi.org/10.1002/2017GL076926, 2018b.